# Marker-Assisted Selection in Breeding for Fruit Trait Improvement: A Review

**DOI:** 10.3390/ijms24108984

**Published:** 2023-05-19

**Authors:** Gloria De Mori, Guido Cipriani

**Affiliations:** Department of Agricultural, Food, Environmental and Animal Sciences, University of Udine, Via delle Scienze 206, 33100 Udine, Italy; guido.cipriani@uniud.it

**Keywords:** marker-assisted selection (MAS), fruit quality, mapping, quantitative trait loci (QTL), Single Nucleotide Polymorphism (SNP), simple sequence repeat (SSR)

## Abstract

Breeding fruit species is time-consuming and expensive. With few exceptions, trees are likely the worst species to work with in terms of genetics and breeding. Most are characterized by large trees, long juvenile periods, and intensive agricultural practice, and environmental variability plays an important role in the heritability evaluations of every single important trait. Although vegetative propagation allows for the production of a significant number of clonal replicates for the evaluation of environmental effects and genotype × environment interactions, the spaces required for plant cultivation and the intensity of work necessary for phenotypic surveys slow down the work of researchers. Fruit breeders are very often interested in fruit traits: size, weight, sugar and acid content, ripening time, fruit storability, and post-harvest practices, among other traits relevant to each individual species. The translation of trait loci and whole-genome sequences into diagnostic genetic markers that are effective and affordable for use by breeders, who must choose genetically superior parents and subsequently choose genetically superior individuals among their progeny, is one of the most difficult tasks still facing tree fruit geneticists. The availability of updated sequencing techniques and powerful software tools offered the opportunity to mine tens of fruit genomes to find out sequence variants potentially useful as molecular markers. This review is devoted to analysing what has been the role of molecular markers in assisting breeders in selection processes, with an emphasis on the fruit traits of the most important fruit crops for which examples of trustworthy molecular markers have been developed, such as the MDo.chr9.4 marker for red skin colour in apples, the *CCD4*-based marker CPRFC1, and LG3_13.146 marker for flesh colour in peaches, papayas, and cherries, respectively.

## 1. Introduction

Fruits play an important role in the human diet. Some have sustained and still somehow sustain populations in parts of the world where other food sources are scarcely available. Staple fruits can provide a large part of starch, proteins, and fats from fresh or processed fruits. Staple fruits include some fruit trees that are of paramount importance in some regions of the world. Among others are breadfruit, plantains and bananas, olive oil, and coconut oil. Modern, more evolved societies see fruits and vegetables as sources of beneficial compounds: they are good for our health. They contain natural active principles, ranging from primary metabolites (nutritive factors, vitamins, and minerals) to secondary metabolites known as phytochemicals [1].

The total world fruit production for 2020 was 887,027,376 metric tons. In 1961, production was 200 million tonnes. The People’s Republic of China, India, and Brazil were found to be the largest producers [2] (FAOSTAT database). Five fruit species accounted for 57 percent of the total production in 2019, down from 63 percent in 2000: bananas and plantains (18 percent), watermelons (11 percent), apples (10 percent), and oranges and grapes (9 percent each) [3] (FAOSTAT 2021 Statistical Yearbook).

Undoubtedly, there are several agronomic traits under genetic control that are important for fruit growers and, therefore, for breeders. We decided to collect in this review the information specifically available on the fruit, not because the other characters are of less interest, but because the fruit well represents the final result of a process that starts from the production of the plant in the nursery arriving on the consumer’s table. Having decided that our focus in this review would be the fruit, we collected the information object of many years of work of the scientific community that a breeder can have at his disposal. We have collected the information in six sections regarding: (1) phenological traits related to fruit production, (2) skin colour, (3) flesh colour, (4) structural qualities, (5) organoleptic qualities, and (6) harvesting maturation and ripening. The goal was to select the molecular markers available for each species and character and, when not available, indicate the path taken and still to be made to identify markers that are useful and efficient in the selection process.

Consumer choice and consumption behaviour, including healthy perception and the pleasure aspect, given the heterogeneity and the multiplicity of the qualitative aspects that can characterise fruit products, have been widely investigated [4]. Sensory factors, absence of defects, quality and safety standards, and environmentally friendly products have become a prerequisite for consumers [4,5,6,7,8]. Focusing on the topics of this review, consumers’ and marketers’ evaluations of sensory factors, including visual appearance, taste, freshness, colour, aroma, texture, shape, crispness, and absence of fruit defects, have to be considered as a guide for breeders. An example of consumer preference evolution, considering both the appearance and the perception of a higher content of healthy compounds, is the fruit skin and flesh colours: plenty of examples are easily visible in all fruit markets. The deep, intense skin colour selected in all fruit species, as in the apple cultivars ‘Fuji’ and ‘Gala’ clones, as well as the yellow and double yellow/red colour of the kiwifruit cultivars (‘Zespri^®^SunGold’, ‘Jingold’, ‘Soreli’, ‘Dorì’, ‘Hongyang’), are just a few examples.

The trait of interest for fruit breeders can be summarised in the following categories: (1) rootstock adaptations to soil pests, diseases, and abiotic stresses; (2) rootstock devoted to minimising the canopy sizes; (3) canopy vigour and adaptation to different training systems; (4) phenology traits (bud break, blooming time, ripening date); (5) pests and disease genetic resistances; (6) fruit size and quality; (7) healthy compounds; (8) fruit resistance to manipulation and post-harvest management.

Considering the fruit, several are the trait of interest: size, shape, weight, skin colour, other skin characteristics (browning, blush), flesh traits (texture, soluble soil content/SSC, sugars, acidity, acids, secondary metabolites), seedlessness, resistance to post-harvest manipulation, and diseases related to the conservation.

Fruit quality traits are indispensable in every fruit breeding program, and selection for other traits has to be considered only from the perspective to add better characteristics in order to get improved cultivars, as well as resistance to pests, for example, avoiding deterioration of quality characteristics. Fruit breeding is a long process due to a usually long juvenility period ranging from one year (strawberry) to decades (avocado, palm date) [9]. Almost all of today’s commercial products reflect the results of continuing breeding programs, and breeders generally lack the capacity to generate new cultivars quickly in response to evolving consumer preferences and crisis situation, such as sudden pests or disease burst, or climate change. The juvenility period is influenced by the environment [10] and inversely correlated to vigour [11]. It is possible to reduce the length of the breeding cycle by manipulating the cultural conditions. In apples, where field-grown seedlings typically do not flower until they are at least 5 years old, plants can be promoted to the adult reproductive phase after as little as 10 months under optimal growth conditions [12]. In recent years, it has become possible to promote flowering using biotechnologies. Nearly 30 years ago, Detlef Weigel and Ove Nilsson showed that flowering could be triggered in aspen by transgenic expression of a gene from *Arabidopsis* called *LEAFY* (*LFY*) [13]. From that pioneering work, scientists quickly move to cultivated species, including fruit species such as apple. The overexpression of *BpMADS4* from silver birch (*Betula pendula* Roth.)*,* a homologous gene of *A. thaliana FRUITFULL* which is another key gene in flower initiation, has been found to induce early flowering in apple plants [14]. By using the *BpMADS4* transgenic apple line T1190 cv. ‘Pinova’, it was shown that one crossbred generation per year is feasible [15]. These biotechnology tools offer the opportunity to speed up the breeding process that could require many generations of crosses and selections, especially when it is necessary to introduce characters from wild genotypes or cultivars of poor commercial value.

Molecular markers made another advance possible: selecting traits for which the phenotype is not easily scorable until late in seedling cultivation, shortening the breeding cycle and reducing costs. With the explosive growth of genetic research and the wide availability of sequenced genomes as a result of the reduction in sequencing costs achieved through second and third-generation sequencing techniques, a huge number of molecular markers are available. Reference genomes are now accessible in a wide number of fruit species, and information is often collected in public databases which may include collections of multiple economically important species belonging to single botanical families or dedicated to single species (https://www.rosaceae.org/, https://kiwifruitgenome.org/, https://banana-genome-hub.southgreen.fr/, https://www.genoscope.cns.fr/externe/GenomeBrowser/Vitis/, accessed on 18 May 2023). Although sequencing costs are low and further reductions can be anticipated in the fairly near future, the availability of sequenced genomes is only the first step in the development of truly efficient molecular markers. Molecular markers can be used for several purposes including, phylogeny, molecular fingerprinting, population studies, and assisted selection. In the latter case, it is essential to associate the markers with a trait under simple, Mendelian, or quantitative genetic control. Association mapping or linkage disequilibrium (LD) mapping [16] or, alternatively, linkage mapping using bi-parental populations [17,18] are usually preferred to screen markers located near the trait of interest, although some simplified methods, such as bulk segregant analysis, have been used successfully (Figure 1) [19].

Quantitative trait locus (QTL) analysis includes a set of statistical methods that links two types of information: phenotypic data (trait measurements) and genotypic data (usually molecular markers), bridging the gap between genes and the phenotypic traits that result from them [20]. Considering that most of the traits of agronomic interest are controlled by many genes, QTL analysis is a prerequisite to identifying the molecular markers associated with those traits.

Marker-assisted selection makes it possible to indirectly identify the presence of the genetic determinant or determinants of the traits of interest in the progeny obtained by crossing the appropriate parents. The success of the MAS depends on several factors, including the complexity of the trait considered and the proximity of the marker to the trait to be selected. MAS is gaining considerable importance as it can improve the efficiency of plant breeding through the precise transfer of genomic regions of interest and acceleration of the recovery of the section of the genome that contains the trait of interest.

Genome-wide selection (GWS) allows for the evaluation of the value of parental potentials in a crossing plan. GWS makes use of genomic estimated breeding values (GEBVs) as selection parameters, rather than the estimated breeding values (EBVs) traditionally used by fruit breeders [21,22]. GEBV is obtained from the phenotypic analysis of a population that is genotyped with a large number of markers, usually using SNPs, to establish the effect of the markers on complex phenotypes, usually controlled by numerous loci. The GEBV of single individuals is calculated on information obtained from molecular markers in order to select the outstanding individuals. These can be used as parents for further generations of crosses or evaluated as cultivars in extensive field analyses. GWS can be made particularly efficient and cost-effective if MAS is used for screening the cross-population with few markers to eliminate unwanted genotypes. This filtering then allows many thousands of markers to be used in order to apply the GWS [23]. The foreground MAS selection in a breeding population for simple ‘must-have traits’, such as pest and disease resistances, flesh or skin colour, rootstock dwarfing ability as in apple and cherry, or gender in dioecious crops such as kiwifruit, enable a substantial reduction in the number of seedlings to be genotyped with dense markers for GWS [24].

It has always been believed that the use of MAS would certainly benefit breeding programs. After all, it seems intuitive that for the characteristics of the fruits, a selection made on the seedlings determines a clear gain in economic terms, considering that a more or less relevant part of the individuals can be discarded. There are not many works that have clearly determined what the genetic and economic gain of the application of MAS could be. A barrier that prevents the diffusion of MAS in breeding programs is the lack of information regarding the cost of applying the technology and the benefits that would derive from it [25].

When a marker can be regularly used in a breeding program for a certain characteristic without additional testing, it is considered verified. The marker’s alleles must be in substantial linkage disequilibrium with the trait locus’s alleles of interest, and they must be consistently associated with measurable positive or negative effects. These are crucial requirements for validation. Such linkage must be demonstrated in parents of a program’s breeding material.

Alternatively, the markers should be placed close to candidate genes in the genome, which means that functional genomics and fine gene mapping are both beneficial for creating viable markers for MAS.

It can be expected, perhaps not surprisingly, that the use of MAS could be distributed, in terms of effectiveness, in a contrasting way considering two important factors, such as the prediction of breeding value and the reduction of costs. Several studies have demonstrated the superiority of MAS in supporting breeding, particularly in academic studies, where technical resources are frequently available [26]. Some studies have examined the economic impacts of MAS in horticultural breeding programs. One study indicated that inheritance of the trait, the timing of trait expression, application timing of DNA testing in a program, and testing costs play important roles in determining cost-efficient MAS [27]. In a breeding program carried out on apple trees, in which the number of seedlings produced each year is very high (30,000), the break-even point (BEP) was evaluated by comparing a traditional selection method with one assisted by molecular markers. The result that emerges clearly shows that an advantage in the use of markers occurred when the rate of removal of seedlings exceeded 13.8%, which constituted the threshold of the BEP [26]. A multi-trait DNA test for apple acidity, crispness, and firmness on both genetic gain and cost efficiency in an apple seedling population has been confirmed compared to phenotypic selection [28]. The MASS (marker-assisted seed selection) efficiency calculator is a spreadsheet developed that allows evaluating the efficiency in economic terms and the effectiveness of molecular analyses in the selection, particularly useful in breeding programs of species with long juvenile periods [29]. The cost reduction threshold has been demonstrated to vary according to the traits considered and the species [30]. It is to be expected that the economic gain will increase with the decrease in the application costs of molecular markers and with the development of tests that allow the simultaneous evaluation of many traits in the breeding populations.

Here, we present a collection of all the available information about the molecular markers that breeders can use to speed up the process of selecting new cultivars of fruit species. We mined the references of the last 20 years (2000–2023) using different databases: Web of Science, CAB Abstract, and Google Scholar, focusing particularly on the breeding concerning the fruit traits. Appendix A summarises the most relevant references on the topic.

## 2. Phenological Traits Related to Fruit Production

Some characteristics, although not directly associated with a fruit, have a strong correlation with it. For example, the budding date, the blooming date, the growth dynamics of the fruit, and the harvest date influence the fruit’s quantitative and qualitative characteristics. Bi-parental mapping and association mapping have been widely used to identify markers associated with phenological traits. QTL analysis for the flowering time allowed for the identification of one significant QTL on the linkage group 5 (G5) in apricot. Two microsatellite loci (UDAp-423r and AMPA-105) (Table 1) were found to be tightly linked (4 cM) to this important agronomic trait [31]. A wide study at the European level used a large panel of apple cultivars genotyped at a high density of SNPs using the Axiom^®^ Apple 480 K SNP array to implement a large Genome-Wide Association Study (GWAS). SNPs were detected for the flowering and ripening period, and genes encoding transcription factors containing either NAC or MADS domains were identified as major candidates within the small confidence intervals computed for the associated genomic regions [32]. Reliable SNPs or QTNs (Quantitative Trait Nucleotides) associated with a blooming date and days after flowering were detected in a GWAS analysis on peach [33]. Those markers were collocated near or in the same regions previously reported using QTL mapping and pedigree-based analysis (PBA) [34,35,36,37]. A comparison between different *Prunus* species was made in order to identify the possible synteny between the species regarding many characters of agronomic interest, including the date of flowering and the length of the period between flowering and harvesting [38]. QTL associated with the flowering period (LG4, LG6) and fruit development period (LG4, LG6) have been identified in peach [39]. In sweet cherry, QTLs were identified in the same chromosomal region (LG4) for two correlated traits, chilling requirement and flowering date [40]. A QTL mapped on linkage group 1 of the early blooming, low chilling requirement Spanish cultivar ‘Cristobalina’ overlapped with dormancy-associated MADS box (*DAM*) genes. This finding is consistent with what was found on peach and indicates that these genes are major determinants of chilling requirements [41]. Most of the candidate genes in the identified regions are involved in chromatin remodelling (*ARP4*, *EMF2*, *PIE1*) and in gibberellin homeostasis (KS and GA2ox). The high synteny existing among *Prunus* species made possible the use of peach and sweet cherry genomes to compare homologous regions such as the QTL on LG4 controlling chilling requirement in peach, sweet cherry, and Japanese apricot [42]. Repeat flowering is an important trait in strawberry breeding programs. One genomic region associated with the total number of weeks of flowering was identified on LG IV-S-1 [43], and two markers could be good candidates for selecting repeat flowering in strawberry. Everbearing, day-neutral, or perpetual flowering (PF) is an important trait in strawberry plants because it influences the flowering behaviour. A large effect QTL for flowering has been detected on LG4A, and the homozygous recessive genotype was highly predictive of seasonal flowering [44]. High-linkage disequilibrium between SNP markers within haploblock 2 of the PF region was detected, and it had a significant association with the flowering habit, although the test accuracy varied among environments [44]. Some preliminary work on QTL mapping has been carried out in several other species: plant height in coffee [45]; vigour, sucker habit, and bud burst in hazelnut [46]; bud burst in pear [47]; pollen sterility in peach [48]; lateral fruitfulness in walnut [49]. Notably, some of the traits have a Mendelian inheritance, such as the pollen sterility in peach (*Ps*). The *Ps* gene was located on the top of the linkage group 6 with the nearest marker being at 4.8 cM (Table 1) [48].

In dioecious fruit species, it is useful to have a molecular marker that allows for the early selection of female individuals. In fact, breeders are generally more interested in selecting productive individuals, and eliminating males saves considerable economic resources. In *Actinidia*, a simple-to-use molecular marker has recently been developed which allows for this type of selection on seedlings [50]. Other sex-linked molecular markers have been developed for various fruit crops such as papaya (SCART1, SCART12, and SCARW11) [51], ginkgo (GBA and GBB) [52], kaki (DlSx-AF4S) [53], mulberry (MBS markers) [54], and pistachio (SNP-PIS-167992 and P-ATL-91951-565) [55].

## 3. Skin Colour

Fruit skin colour represents one of the wide ranges of desirable characteristics that determine the quality of fruit. This trait is important for both the commercial and aesthetic value of fruit; indeed, people are generally drawn to an intensive and broadly covered colour of fruit skin (Figure 2). Moreover, in some cases, such as the russet skin of Japanese pear, the skin colour also represents a protective barrier against external stress caused by disease, insects, bad weather, and shipping [56]. The trait in several fruits is primarily determined by the content and composition of anthocyanins, which are the largest and most diverse group of water-soluble phenolic plant pigments derived from the phenyl propanoid pathway. During the last decade, many structural genes involved in the anthocyanin biosynthetic pathway and various transcription factors (*MYB*, *bHLH*, and *WD40*) have been identified and functionally characterised in fruit crops [57].

The biosynthesis of anthocyanin in many plants is affected by the temperature, which plays an important role in the regulation of these pigments’ accumulation. Several studies [58,59,60] describe how high temperature causes poor fruit skin colouration, especially on early harvested cultivars. The prediction of colour by the detection of marker genes before the tree even starts to bear fruit and the development of new colour fruit cultivars that are adapted to a warm climate would therefore be of much practical utility.

High-resolution QTL mapping was used to detect two main colour-related QTLs located on the LG3A linkage group in Octoploid Strawberry (*Fragraria* × *ananassa*). Moreover, Labadie and co-workers [61], through genetic mapping, trascriptome analysis, and whole-genome sequencing, enabled the detection of a homoeo-allelic variant of the anthocyanidin reductase (ANR) enzyme, which is involved in the control of anthocyanin accumulation. The authors identified a deletion of 18 bp in the 5′UTR of the *ANR* gene that underlies a major colour QTL on LG LG3A and designed a simple marker that can be used as a genetic marker for accelerating the selection process of a more intense red colour strawberry (Table 1).

As stated above, anthocyanin biosynthesis involves a wide range of constructive genes, including anthocyanidin synthase, that are regulated by *MYB* transcription factors. These genes are co-ordinately expressed, and their levels of expression are positively related to the anthocyanin concentrations [57]. Consequently, expression or suppression of the constructive genes contributes to a variety of changes that make fruits visibly different from the colour of skin point of view. The quantitative effect of *MYB*-related genes on anthocyanin accumulation was reported for several fruit crops (e.g., grapevine, apple, peach, strawberry). The development of a simple sequence repeat marker associated with the gene *MdMYB1-1* called Mdo.chr9.4 could be a good tool for red skin colour selection in apple [62]. This marker shows polymorphism amplifying of four alleles (Mdo.chr9.4-R0, Mdo.chr9.4-Y_3, Mdo.chr9.4-Y_9 and Mdo.chr9.4-Y_15) distinguished by length and was mapped at the bottom of the LG9 of apple genome version 1 [63] on which *MDMYB1-1* were located. In major cultivars and founders used in the study, the red skin colour phenotype was only associated with the 157 bp allele designed as Mdo.chr9.4-R0. The other alleles are 154, 148, and 142 bp, designed, respectively, as Mdo.chr9.4-Y_3, Mdo.chr9.4-Y_9, and Mdo.chr9.4-Y_15, and they were associated with yellow skin colour [62]. The GWAS study permitted for the dosing of the quantitative effect of the *MDMYB1-1* gene, revealing that this gene was the major and principal genetic factor, not only as a red skin colour determinant, but also of the red intensity variation in apples, and the selection of Mdo.chr9.4-R0 dose 2 individuals was expected to result in the selection of individuals with an intense red colouration. However, it has been revealed how epigenetic mechanisms affect *MDMYB1* gene expression and skin colour variation in ‘Gala’ [64]. Moreover, there is evidence that the red skin colour of the ‘Gala’ apple cultivar is also affected when grown under challenging conditions where summer temperatures are high [65]. Chagné and co-workers [66] aimed to provide a new robust genetic marker for red fruit skin colour that can be applied by breeders for the selection of seedlings that will express a high-red colour, even when grown under high temperatures, wherein they mapped the locus associated with red skin colour using a large set of seedlings and developed a robust allele-specific co-dominant PCR-based marker for the locus and validated it in several cultivars and breeding populations grown under warm summer conditions. The authors used an IRSC apple 8-K SNP array to identify the most significant SNP marker (ss475879531) that was then transformed in a marker suitable for use in a real-time PCR assay. The individuals that were homozygous for the red allele (GG) using the ss475879531 marker (Table 1) consistently bore fruit with high-red intensity, high-percentage overcolour, and high-anthocyanin concentration [66]. Chagné and co-workers [67] also designed the apple International RosBREED SNP Consortium OpenArray v1.0 (IRSCOA v1.0) assay using a set of 128 apple SNPs linked to fruit quality and pest and disease resistance trait loci and they identified nine SNPs that were independently associated with fruit skin colouration, providing additional markers around the *MYB10* candidate gene.

Genotyping by Sequencing (GBS) for SNP-based linkage analysis allowed for the identification of genomic regions significantly associated with fruit skin colour (LG3 and LG4 of both parents ‘98-99′ and ‘Angeleno’ and both years of observation) also in Japanese plum (*Prunus salicina* Lindl.) [68]. The authors identified a SNP, S3-12879559, in the LG3 that is colocalising close to a *MYB* transcription factor in their populations. In *Prunus salicina,* Gonzales et al. [69] also developed several EST-SSR markers from genes coding for putative flavonoid pathway transcription factors *PsMYB10*, *PsMYB1*, and *PbsHLH35*. Three EST-SSRs identified a different pattern of grouping than that obtained by SNP loci alone. The three EST-SSRs showed three groups, one of which was composed mainly of yellow-skinned cultivars, thus having the potential to be used as molecular markers to be evaluated in MAS strategies [69].

Anthocyanins are responsible also for the red over-colour in peach, while the background is determined by the carotenoid content of the flesh and is controlled by a single locus in LG1 of the peach genome [70,71]. Several pieces of evidence reported that the degree of the peach skin’s red blush is a trait controlled by multiple genes [72,73,74], but in other cases, the red skin colour character has been described as a single Mendelian trait. Beckman et al. [75] observed that the non-red skin colour called the highlighter (H) phenotype trait segregates as a single gene (*H*/*h*) that, once in recessive form (*hh*), is responsible for the highlighter phenotype, and Tuan et al. [76] demonstrated that a *MYB10* gene (*PpMYB10.1*), part of a cluster of three *MYB* transcription factors located on LG3, was involved in the control of the skin colour phenotype in two peach cultivars and developed a marker based on an indel polymorphism of this gene sequence that discriminated for red and non-red skin colour in at least in 23 Japanese cultivars. Bretò and co-workers [77] analysed an F2 population of 276 individuals segregating for the H phenotype and mapped the *H* gene on LG3 of the peach genome, observing that the marker based on *PpMYB10.1* co-segregated with the trait. The *PpMYB10.1* marker presented two alleles of 426 and 609 bp, associated with the *h* and *H* alleles, respectively (Table 1). However, the authors observed that the prediction value of the skin colour phenotype based on this marker was partial, meaning that some of the considered cultivars had different phenotypes to those predicted [77], confirming that it could be used as an efficient screening marker but not from the diagnostic point of view.

QTL analysis for skin-related traits was also performed in cashew apple, and two markers were identified with significant association with QTLs for red intensity (RI), three for brightness (BR), and three for yellow intensity (YI) [78], but QTLs need to be validated before the use in breeding programs.

A GWAS study of fruit-quality traits allowed for the identification of one QTL on the scaffold number 3 of the clementine reference genome v. 1.0 (http://www.phytozome.net, accessed on 18 May 2023) for fruit skin colour also in citrus [79]. Within citrus species, Yu and co-workers [80] reported the construction of genetic maps using a mandarin F1 population of ‘Fortune’ x ‘Murcott’ and identified a cluster of QTLs for flavedo colour and juice colour in a single genomic region on LG4. Carotenoids are described as the major chemical components involved in the determination of mandarin fruit colours, and the authors detected two carotenoid biosynthetic pathway genes, *pds1* and *ccd4*, within the QTL interval associated with flavedo colour, with several SNPs identified within the region being potentially useful in MAS. For instance, the QTL FCAB8 linked marker m1336_s8 could be used to screen individuals for external colour traits, as individuals with allelic configuration BB usually exhibited more orange colour relative to individuals with allelic configuration AB [80]. However, also in this case, QTLs should be validated.

A major QTL associated with skin colour was detected also in Japanese pear (*Pyrus pyrifolia* Nakai) at 2,2 cM on the top of LG8 of the ‘Akikari’ parental, an early maturing russet-skin type cultivar. In this case, fruit skin colour was classified into five classes according to the suberin formation area on the surface of mature fruit [81]. An SSR marker called Mdo.chr8.10 (GA)_11_, newly developed by the authors, was found near the detected QTL, as well as another marker (OPH-19_425_) linked to a gene controlling fruit skin colour previously identified by Inoue et al. [56] using bulked segregant analysis of two F1 progenies of Japanese pear cultivars. The identified QTL would be useful for MAS in Japanese pear breeding programs but, even in this case, further analysis is required.

## 4. Flesh Colour

Colour of fruit flesh is another important quality parameter that influences consumer acceptance. The appearance of the fruit products is the first important information that customers obtain. Furthermore, pulp colour is a fruit parameter that is widely used during the harvest and post-harvest stages to estimate ripeness and maturity, as well as an indicator of flavour, nutritional value, and defects. The main factors affecting pulp colour are the higher concentrations of carotenoids and anthocyanins, as well as the quick breakdown of chlorophyll.

In *Prunus* species, several QTLs were found associated with flesh colour. As previously mentioned, it is well known that the ground peach flesh colour is controlled by a single locus named *Y* mapping to the LG1, with the white flesh phenotype dominant over the yellow one [70,82]. Brandi and co-workers [83] proposed a gene encoding a carotenoid cleavage dioxygenase (*CCD*) to be the major factor responsible for carotenoid degradation in white peaches. Falchi et al. [71] described the association between the locus encoding the *CCD4* (*PpCCD4*) localised in LG1 and the flesh colour for 37 peach varieties, including two somatic revertants and three ancestral relatives of peaches, providing definitive evidence that this locus is responsible for the flesh colour phenotype. The authors show how yellow peach alleles arise from ancestral haplotypes later after at least three independent mutational events involving nucleotide substitutions, small insertions, and transposable element insertions. Similar results were described by Adami and co-workers [84], who developed three distinct types of markers (SSR, SCAR, and SNP) to distinguish *CCD4* alleles associated with the background colour of peaches using two mutant systems, a yellow/white segregating progeny and more than 100 different genotypes representative of the peach germplasm variability. A microsatellite within the *CCD4* sequence that was analysed using a specific primer pair (*ccd4*-FL-f/*ccd4*-FL-r) developed by the authors assisted in the identification of the alleles y^1^ (for the (CT)_9_ allele) present in both the yellow-flesh colour ‘Redhaven’ cultivar and its white-flesh mutant ‘White ‘Redhaven’, and in the case of the yellow cultivar, an allele named y2 (for the (CT)_8_ allele) showed a 6263 bp insertion within the intron with two 490 bp direct repeats at the extremities, similar to an LTR retrotransposon. Furthermore, an allele identical to y2 but lacking the insertion in ‘White Redhaven’ that was putatively associated with the white phenotype was named W^1^. A specific marker (primer pairs: *ccd4*-INS1-f/*ccd4*-E2-r) was designed by the authors to rapidly detect the insertion (Table 1). The presence of alleles *y1* and *W^1^* was confirmed also in another white-fleshed cultivar known as ‘Caldesi 2000′. Selection for the insertion within more than 100 genotypes has enabled the ability to identify all cultivars bearing at least one copy of *ccd4*-*W^1^* as a white-flesh phenotype cultivar, with the exception of ‘Gialla Tardiva’, ‘Jonia’, and ‘OroA’, that despite being yellow-fleshed, possess the allele *W^1^* characterised by a single base substitution (A to T) within the coding sequence that results in a truncated protein. A specific TSP-SNP marker assay was developed by Adamy et al. [84] to analyse this third allele called *y^3^*.

QTLs for background flesh colour were also identified in apricot (*Prunus Armeniaca* L.) in LG1 and LG6 [85]. The first QTL showed good stability during 3 years of observation and is flanked by SSR markers UDAP-441 and PGS1.21. However, they are seldom suitable for MAS without further testing and validation.

Carotenoid pigments are the main players in the flesh colour regulation and also in several tropical fruits (i.e., papaya, guava, mango). Especially with regard to papaya fruit flesh colour, the composition in terms of vitamin A and different carotenoids drew the attention of breeders and consumers. Papaya fruits can be divided into red and yellow flesh fruit. The difference is due to the accumulation of lycopene in red flesh and β-carotenoids in yellow flesh. From the market point of view, red flesh fruits are characterised by a rapid softening and have a shorter shelf life. The phenotypic variation between red and yellow colour is controlled by a di-nucleotide insertion mutation in the gene encoding the enzyme lycopene β-cyclase (*CpCYC-b*). A codominant marker called CPFC1 (*C. papaya flesh color 1*) was developed on the basis of a 36 bp indel detected in the region downstream of the *CpCYC-b* coding region (Table 1) [86] . Vázquez Calderón et al. [87] tested the CPFC1 marker in genotypes derived from the hybrid cross of ‘Marandol’ x ‘creole’ papaya. Three allelic forms were observed in the evaluated progeny: a single 600 bp band (homozygous recessive allelic condition) linked with reddish-orange pulp, dual 660 bp and 550 bp bands linked with yellow flesh colour, and a single 550 bp band (homozygous dominant allelic condition) linked with the light-yellow flesh colour shown by a wild material. The authors observed that the classification based on the molecular screening carried out in young plants matched with the morphological analysis conducted between 7 to 9 months later and suggested that CPFC1 could be a good candidate marker in MAS selection versus red flesh papaya, although it cannot discriminate between the range of different yellow nuances. For the selection of pulp colour in guava fruit, sequence-specific amplified polymorphism (SSAP) and SSR were developed. Clustering of SSAP results and the analyses with the SSR mPgCIR161 suggested that both types of markers could be potentially useful in the selection of specific pulp colour in guava and could be used in the early selection of elite material [88].

Aiming at human health, within the varietal innovation programs, the attention shifted to anthocyanin-rich fruits. Indeed, anthocyanins, thanks to their antioxidant properties and possible involvement in reducing the incidence of certain important human diseases (i.e., cancers, coronary heart diseases) and oxidative stress, as well as their role in preventing age-related declines [89,90,91], appeal to the public and the research community. Some cultivars of peach exhibit an intensely pigmented fruit mesocarp (Figure 3) with a red-violet colouration, mentioned as blood-flesh peach, and two different loci were found associated with this kind of phenotype. In the case of ‘Harrow blood’ cultivar, this character is controlled by a single recessive locus *bf* (*blood-flesh*) that was first described by Warner et al. [92] and then mapped to the top of the LG4 by Gillen and Bliss [93]. On the other hand, in the case of the Chinese peach cultivar ‘Wu Yue Xian’, the blood flesh phenotype is controlled by the dominant locus *DBF* (*Dominant Blood-Flesh*) mapped to the top of the LG5 [94]. Three genes spanning the region of the QTL were purposed as candidate genes for the control of the trait in the latter case. These genes cluster are members of the dihydroflavono-4-reductase (*DFR*) gene family involved in the biosynthesis of anthocyanins. The *DBF* locus is flanked by two SSR markers, AMPP157 and AMPPG178, that provide a good basis for the MAS of favourable individuals at the plantlet stage [94]. A single dominant gene, *DBF2*, was also described and mapped using two interspecific almond x peach populations derived from the cross between ‘Texas’ almond and ‘Earlygold’ peach [95]. The *DBF2* dominant allele associated with the red flesh colour was mapped even in this case on the LG1 between 41,709,139 and 42,754,924 bp of the v2.0.a1 version of the peach genome assembly and the closet marker, snp_1_41535285, is 0.1 cM from the gene. Moreover, Donoso and co-workers [95] detected also a QTL for the fruit flesh colour (*qBF3*) in the central region of LG3 in TXE.

On the basis of the anthocyanin colouration of the flesh, peach cultivars with red colour around the stone and cultivars with red dots present in the flesh can be distinguished. In the case of the first trait, a dominant allele of the gene called *Cs* results in a coloured phenotype. That gene was localised on LG3 through a segregating population and GWAS analysis [96,97]. A deletion occurring in the promoter region of the gene *PpMYB10.1* was found to be associated with the trait [98]. More recently, Zaracho et al. [99] focused their attention on the study of this trait identifying a major QTL explaining 24,4% of the phenotypic variance in LG4 in both intraspecific and interspecific populations between almond and peach. Two SNP markers flank the QTL (SNP_IGA_409167 and SNP_IGA_419762), which corresponds to the physical position between 10,363,642 and 13,675,935 bp in *Prunus* reference genome v2.0. However, the authors found a major gene for maturity date (*MD*) located in the region spanning the QTL and speculated about a possible interaction within the allele from *MD* that they found in LG4 and the *Cs* allele from LG3 producing a red colour around the stone phenotype. Red dots in the flesh phenotype were also investigated. The trait has been mapped as a major QTL called *qRDF5* at the beginning of LG5 in a region between 0.6 and 1.4 Mb of the *Prunus* reference genome v2.0. As stated above, within the concerned region, the *DBF* gene was identified, so a possible explanation speculated by the authors [99] is that *qRDF5* is produced by an alternative allele of the *DBF* gene.

A major QTL for flesh colour was identified also on LG3 in sweet cherry fruit (*Prunus avium* L.) [100]. In this case, the results suggest that the *PavMYB10* gene may underlie the major QTL for cherry fruit colour and control up to 94% of the trait variation with red dominant to yellow colour, but epistatic interactions of this QTL with two minor QTLs in the LG6 and 8 may define the range of colour phenotypes. A targeted approach based on the hypothesis that *MYB10* is the major determinant of flesh colour was used by Stegmeir and co-workers [101] to develop a DNA test for fruit flesh colour in tetraploid sour cherry. Actually, with diploid sweet cherry being one of the progenitor species of tetraploid sour cherry, the authors assumed that this locus could be exploited to develop a predictive DNA test for dark purple-red flesh colour, aiming at the negative selection against the trait and the development of new improved ‘Montmorency’ cherry types that are characterised by the hallmarks of yellow flesh and brilliant red skin colour, which is preferred in the U.S. markets. The cherry 6K Infinium^®^ II SNP array was used to investigate the *MYB10* region, and six haplotypes were found to be significantly associated with variation in flesh colour, providing new insights into the role of the gene in the trait control. The presence of one of these haplotypes (D1) was significantly associated with the darkest flesh colour class, and a diagnostic SSR primer pair called LG3_13.146 (Table 1) [101] was able to uniquely identify D1. This feature makes it an excellent candidate for MAS breeding, both for the negative selection of dark purple-red fruit flesh colour as in this particular case, and for positive selection in other kinds of markets (i.e., the European market), where intense colouration is preferred for cherry juice and other products.

## 5. Structural Qualities

### 5.1. Fruit Size and Weight

Rapid morphological differentiation between closely related organisms is the result of strong selection pressure. About 10,000 years ago, with the so-called Neolithic revolution, humanity began to pass from a society made up of nomadic tribes of hunters and gatherers to a sedentary one in which agriculture developed, accompanied by plant and animal domestication. The selection process of those early farmers was aimed at traits such as fruit size and weight. The cultivated species, specifically fruit and vegetables, appeared morphologically different with much larger fruits and higher productivity than the wild ancestral species [102]. Fruits are characterised by different shapes and sizes, but each fruit results from the development of a single growing meristematic point, and the final size depends on the number of cell divisions, plans of division, and cell expansion during the development of the ovary and of the fruit. Individual fruit shape and size are determined by the genetics, phytohormones, and environment of the fruit. Understanding the genetic control of fruit size and weight is an important goal of breeding projects, as these traits are of primary importance for the fresh market. Given their importance, these characters have been the subject of numerous studies also in fruit species. They are quantitative characters studied through mapping in biparental populations or with association maps. The markers used are mainly SNPs more or less associated with the phenotypic characters, and the genes putatively involved in genetic determinism have been sometimes identified (Table 2). Although the domestication of cultivated plants has led to the selection of genotypes with larger fruit sizes in numerous families, including that of *Rosaceae*, the understanding of the genetic changes that resulted in this fruit size increase between domesticates and their small-fruited wild relatives is most advanced in tomato [103]. A family of genes, named *CNR* (*Cell Number Regulators*), are involved in cell proliferation and fruit size. *FW2.2* is a member of *CNR* first identified in corn [104]. Twenty-three genes of the *FW2.2/CNR* family have been identified in the peach genome. Two were located within confidence intervals of major quantitative trait loci (QTL) previously discovered on linkage groups 2 and 6 in sweet cherry (*Prunus avium*), named *PavCNR12* and *PavCNR20*, respectively (Table 1). Considerable evidence based on sequence analysis, segregation, and association with fruit size was found regarding the role of *PavCNR12* in the sweet cherry linkage group 2 fruit size QTL [105]. In Chinese cherry, an association mapping study with SSR identified three loci associated with fruit size. The three candidate genes thought to be most likely involved in the expression of the trait have been located (scaffold_8_1113, 1115, and 1116) on LG8, which could respond to auxin; scaffold_2_3446, involved in the regulation of cell differentiation; and scaffold_3_269, involved in pectin biosynthetic process [106]. QTLs were mapped for the single-nut weight (SNW), nut width (NW), nut thickness (NT), and nut height (NH) in Chinese chestnut. One pleiotropic QTL at 23.97 cM on LG I might simultaneously control SNW, NT, and NW. The major QTLs were related to the NT [107]. In Citrus, four significant QTLs for fruit weight were identified using GWAS analysis, in part confirming previous reports [79]. Major QTLs (R^2^ ≥ 30%) were detected for equatorial diameter [108]. A QTL for yield was detected on LG2 in coffee, and it was significant in two different locations. The positive effect on yield was always associated with the allele inherited from the genotype ‘CCC1146′ at this locus. Two QTLs were detected for bean size using single-point analysis. They were located on different LGs (4, 6) and were significant in only one environment each [45]. 1 REMAP locus (Gret1Fb-816-2), a retrotransposon-microsatellite-amplified polymorphism, was associated with berry weight in grape [109]. For the same trait, five new QTLs, on linkage groups (LGs) 1, 8, 11, 17, and 18, were identified in addition to the known major QTL on LG18. The QTL with the largest effect explaining up to 31% of the total variance was found on LG17, where it colocalised with a published putative domestication locus. This locus was also the most stable over time. Other loci were less stable in the population, and this was consistent with the highly composite nature of berry weight, affected by numerous factors (cell multiplication, cell wall modifications, photosynthesis, sucrose, water transport, growth regulators, etc.) and therefore expected to be under polygenic control, with different causal polymorphisms segregating in different populations [110]. One major QTL identified in LG11 was stable in the four years of the field experiment. The QTL peak was always close to or coincided with marker VVIB19 (Table 1). The candidate genes in the region were an auxin-responsive protein *IAA9*-like (*Aux/IAA9*) gene and a DELLA protein *SLR1*-like gene. These genes are likely to contribute to auxin and gibberellin signalling, respectively, which affect fruit development and enlargement [111]. In Japanese plum, the most significant QTLs for fruit weight were found in the LG2. The SNP marker 2_18489481 was the most significant cofactor, with an additive effect of 5.5 g. The closest SSR to this QTL was UDAp456 (Table 1), where individuals bearing the ‘f’ allele had fruits with around 13 g more than other genotypes. In ‘Angeleno’, a minor but significant QTL was detected on LG7 (cofactor 7_20071259, an additive effect of 4.2 g) [112]. The fruit species grown in tropical and sub-tropical countries have been subject to less consideration by scholars due to the lower economic resources available in the countries of origin. However, following the improved economic conditions and the lowering of the costs of molecular analyses, numerous studies have been published recently. Recent work on the longan has highlighted the presence of QTLs for some pomological characters, including fruit weight. Twelve stable QTLs for single fruit weight (SFW) were identified on LG1, LG4, LG10, and LG14. Three verified genes belonging to three stable QTLs were associated with the SFW trait. An *FW2.2/CNR-like* gene *DlCNR8* (*Dlo_011045.1*) was identified on LG10. The expression of *Dlo_011045.1* was significantly increased during the early fruit development stage in the small fruit progeny line, while no expression change was observed in the fruit development stage of the large fruit progeny line, suggesting that *DlCNR8* (*Dlo_011045.1*) negatively regulates fruit development. The other two candidate genes identified were a gene encoding cytochrome *P450* (*Dlo_034894.1*), a member of CYP78A that plays important roles in regulating organ size or weight by affecting cell division, and *EXP4* (*Dlo_007693.1*), an expansin involved in many plant growth and development processes, and they both affect organ size and yield [113]. Quantitative trait loci (QTLs) associated with guava fruit traits such as fruit length, fruit width, fruit weight, seed weight, inner pulp thickness, and yield were located among the 11 linkage groups representing the haploid genome number of guava. QTLs were identified in papaya for fruit weight, fruit length, and fruit width. Fruit filling in banana was found to be associated with multiple major QTLs and bunch weight, and its related traits were associated with chromosome 3 [114].

QTL regions associated with peach fruit diameter and weight have been detected in all chromosomes [39,115,116,117,118,119,120]. Four QTLs were identified for the fresh weight (FW), two located on LG4 and one on LG5 and LG6, and the other four for fruit diameter (FD), two located on LG4 and two on LG5, in peach. The 130 identified candidate genes for FW and FD completely overlap [120]. Da Silva Linge et al. [33] found two reliable QTLs associated with FD on LG1 and LG7 and five QTLs associated with FW on LG1, LG2, LG3, LG4, and LG6. Positions of the QTLs were compared with previous reports. The qtnFDIA_7.1 on chromosome 7 is in the vicinity of fruit width and fruit depth QTLs (qP-Fwd7.2 and qP-Fd7.2) reported by da Silva Linge et al. [115]. The qtnFDIA_1.1 was identified in a different region of chromosome 1 when compared with previous linkage analyses [115].

Marker trait associations (MTAs) were identified for the 25 traits studied in walnut with GWAS analysis, including weight and size. MTAs were found on chromosomes (Chr) 5, 8, 11, and 15 for nut length; on Chr6, 7, 9, 14 for nut volume; and on Chr1, 3, 6, 8, and 12 for nut weight. The association on Chr14 controlled three different dimensions of the nut (nut face diameter, nut volume, and kernel volume). Minor but significant associations were found that could have some implications for selection: the authors suggest it could be possible to select superior genotypes for larger nut size in general (MTA on Chr14), but it remains possible to select for bigger kernel volume using only minor loci. A candidate gene coding for a beta-galactosidase was found to be associated with nut face diameter, nut volume, and kernel volume. A receptor-like cytosolic serine/threonine protein kinase *RBK1*-encoding gene for nut length and a *LAla*- D/L-amino acid epimerase-encoding gene associated with nut surface area and nut volume were found. Finally, a chaperonin 60 subunit alpha 2, chloroplastic gene, and a *BEL1*-like homeodomain-protein-4-encoding gene were found to be associated, respectively, with nut weight and kernel weight [121].

### 5.2. Seedlessness

Since seedless fruit are favoured for fresh eating and processing, seedlessness is a key area of research in grapes. The mapping and identification of genetic loci influencing this feature have received much attention in the quest to create molecular markers that can speed up and improve seedless grape breeding. At least seven loci have been reported to map to the quantitative trait of seedlessness [122,123,124], and several molecular markers related to seedlessness have been developed, including the random amplified polymorphic DNA (RAPD) marker UBC-269500 (Table 1) [125], the sequence characterised amplified region (SCAR) markers SCC8 (Table 1) [126,127] and SCF27 (Table 1) [128], and quantitative trait locus (QTL) linked to the marker VMC7F2 (Table 1) [124]. Through a combination of transcriptome and genome analysis in two F1 mapping populations, according to Royo et al. [129], the *MADS*-box gene *VviAGL11* missense mutation (Table 1) is the primary cause of seedless grapes. The *VviAGLl1* gene of stenospermocarpic grapes contains a single-nucleotide mutation that results in the substitution of arginine-197 to leucine [129]. When screening for the seedless trait, marker effectiveness depends on the genetic background. The p3_*VvAGL11* marker can accurately identify seedlessness in about 85 percent of the offspring when the lignified seed is lacking in seedless grapes [130]. Wang et al. [131] identified on chromosome 13 a SNP of the gene G8, in position 11,240,814 (variant G/A) (Table 1), closely related to grape seedlessness that showed 67.5% efficiency in distinguishing the seeded and seedless progeny.

Seedlessness is also an important trait in citrus breeding programs. According to Yamasaki et al. [132], parthenocarpy, early halted seed following fertilisation, male or female infertility, and self-incompatibility can all result in citrus seedlessness. Similarly, Gmitter et al. [133] explained how ploidy manipulation could create seedless triploid hybrids by mating tetraploid and diploid parents. Bulked segregant analysis was used to identify markers associated with the seedless locus in *Citrus kinokuni* [134]. Four RAPD primers flanking the seedless locus were mapped (Table 1) close to the seedless trait: OPAI11-0.8 at 8.7 cM, OPAJ19-1.0 at 8.4 cM, OPM06r-0.85 at 4.3 cM, and OPAJ04r-0.6 at 6.4 cM [134]. Using the amplified fragment length polymorphism (AFLP) method, the Ponkan mandarin (*Citrus reticulata* Blanco) was investigated for molecular markers associated with the seedless phenotype. Successful SCAR (sequence characterised amplified region) markers were developed from AFLP-2 and AFLP-5 amplicon sequencing (Table 1) [135]. Seedlessness in Satsuma (*Citrus unshiu* Marcov.) is caused by a combination of male sterility, female sterility, and parthenocarpy. ‘Okitsu No. 46’ and ‘Okitsu No. 56’ (O46–O56) crosses were used to map QTLs for the traits apparent pollen fertility (APF) and number of pollen grains per anther (NPG), which are utilised as indicators of male sterility. On linkage group 8, a putative QTL for NPG (MS-P1) was found with a significant LOD score (7.31) and 47% of variation explained. On linkage group 6, a QTL for APF (MS-F1) was found with a significant LOD score (5.71) and 63.6% of variation explained. In cross populations of kunenbo progenies, male sterile seedlings can be selected by employing neighbouring SSR markers related to MS-P1 and MS-F1 haplotype blocks. To confirm a more widespread application of the markers, additional testing must be carried out utilising additional F1 populations and Satsuma cultivars with a wide range of progeny [136].

Flowers on *Diospyros kaki* are unisexual. The varieties bear seeds when the male flowers are present. Pollination constant astringent (PCA), pollination variation non-astringent (PVNA), and pollination variant astringent (PVA) types with seeds produced unmarketable fruits. DlSx-AF4 (Table 1) [53] is a marker developed to select for genotypes with male flowers. This marker, generated in the *D. lotus* by bulk segregant analysis (BSA) and amplified fragment length polymorphism (AFLP) marker, has been confirmed in a persimmon germplasm collection, and the results obtained were in agreement with the phenotype data. Because they stop accumulating tannins during the early stages of fruit development, PCNA (pollination constant non-astringent) cultivars are greatly desired because their mature fruits are not astringent. In Japanese varieties, the PCNA trait is recessive to the non-PCNA trait and is controlled by a single locus, AST. The area related to AST has been the focus of numerous efforts. The most encouraging findings came from a study that pinpointed an area that was closely connected to the AST gene [137]. On the basis of primers designed from the identified region, these authors developed a very reliable multiplex PCR approach that allowed for the detection of recessive and dominant alleles. A number of variants have been tested using these primers [138]. Microsatellites in the region enable the separation of 12 different alleles from 14 non-PCNA genotypes. Blasco et al. [139] validated both the DlSx-AF4 and the microsatellite set linked to the locus AST in 42 cultivars widespread in Italy, Spain, and Japan.

### 5.3. Firmness, Crispness, Texture

In fleshy fruits, softening is a sign of ripening and can have both positive and negative effects on postharvest stability and texture. In order to increase fruit quality through breeding, it is important to focus on the timing and degree of pre-harvest ripening as well as the accompanying textural changes after harvest.

Fruit ripening is a complicated process that involves the development of flavour, colour, cell wall changes and softening, starch breakdown, and aroma, all of which are significant contributors to the distinctive fruit qualities. The process is controlled at many levels by a complicated network of interactions between the plant hormones and essential transcription factors [114].

Apple fruits are known for their long storage life, although a wide range of flesh softening occurs among cultivars. Loss of firmness is genetically coordinated by the action of several cell wall enzymes, including polygalacturonase (PG), which depolymerises cell wall pectin. A single-nucleotide polymorphism (Md-PG1) (Table 1) was positioned in the linkage group 10, co-located with a quantitative trait locus (QTL) identified for fruit firmness in post-harvest ripening. Fruit firmness and softening analysed in different stages, from harvest to post-storage, determined a shift of the QTL from the top of this linkage group to the bottom, where *Md-ACO1*, a gene involved in ethylene biosynthesis in apple, is mapped. This PG–ethylene-related gene has been positioned in the apple genome on chromosome 10, which contains several QTLs controlling fruit firmness and softening [140]. 

Another simple sequence repeat marker, Md-PG1SSR 10kd, suitable for the selection of high-texture performance seedlings, was identified (Table 1) [141]. A large effect locus associated with the harvest date and firmness using genome-wide association studies (GWAS) was identified. Polymorphisms in or around a transcription factor gene, *NAC18.1*, could cause variation in these traits. Comparing a genetic marker within *NAC18.1* to markers targeting three other firmness-related genes widely used by breeders (*ACS1*, *ACO*1, and *PG1*), strong evidence was found that the *NAC18.1* marker was the strongest predictor of both firmness at harvest and firmness after 3 months of cold storage (Table 1). By sequencing *NAC18.1* across 18 accessions, two predominant haplotypes containing the single-nucleotide polymorphism (SNP) previously identified using GWAS, as well as dozens of additional SNPs and indels in both the coding and promoter sequences were found. *NAC18.1* encodes a protein that is orthogolous to the NON-RIPENING (NOR) transcription factor, a regulator of ripening in tomato (*Solanum lycopersicum*) [142]. Four SNPs are suggested for MAS for firmness and crispness: one SNP on LG15 (AHZAHRE targeting *MdACS1*) and one of three SNPs on LG10 (AHKA3OR, AHPAWDN, or AHMSZ07) targeting *MdPG1* [67].

The two QTLs of fruit firmness (FF)F were located on LG1 and LG2 of a maternal parent in a bi-parental cross of apricot. A total of 117 candidate genes for FF were identified in two intervals. These candidate genes were involved in processes such as cell metabolism, cell wall composition or degradation, vesicular transport, transcription factors, and signal transduction [143].

Pulp texture—mostly firmness and melting—is the second most important factor regarding the eating quality of bananas. For pulp firmness (PF), QTLs were detected on LG1, 3, 4, 5, 6, and 11. Seven QTLs associated with PF contained candidate genes for cell wall metabolism (division and expansion) and hormones controlling fruit ripening. The majority of these genes, including *PFWQTL-PM11*, *PFWQTL-G6*, and *MQTL-G4.1*, were involved in the control of auxin and peroxidase activity (1 Mb). Moreover, *PFWQTL-G11* candidate genes for the ethylene response (ETR) were found (1.5 Mb). In QTLs associated with numerous quality traits, including PF, genes involved in cell wall metabolism were discovered. Three of these nine QTLs—pHWQTL-PM1 7.1, DMCWQTL-PM9.2, and PFWQTL-G6—corresponded to large consensus QTLs with R^2^ values of more than 25.0%. Before considering MAS selection, they need now be confirmed in other genetic situations, and their confidence intervals would need to be improved [144].

In grape, two QTLs for firmness were located on the LG3 (Iku82 and consensus maps) and LG10 (Iku82 map) bi-parental map [111]. Both QTLs were confirmed in the 4-year average. The QTL peak on LG3 was always located close to or coincided with the marker VMC2E7 (Table 1). The QTL on LG10 was located in the top to the middle region near marker VVIH01 or UDV073. The position of the QTL on LG10 varied over years, whereas the QTL on LG3 tended to be more stable. Carreño et al. [145] found QTLs for firmness on LGs 1, 4, 5, 9, 10 (bottom region), 13, and 18 [111]. Genetic mapping and quantitative trait loci (QTL) research revealed that linkage groups 8 and 18 include the majority of the determinants for firmness in table grape. The largest contributor to this QTL is a male allelic additive effect. With confidence intervals up to 10 cM, these two QTLs accounted for around 27.6% of the phenotypic variance together. A cation/calcium exchanger, a xylosyltransferase, a possible cellulose synthase, and a potential invertase among the dozens of genes identified in these two QTLs were considered to be the most important candidate genes for this trait [146]. By employing composite interval mapping, three potentially significant quantitative trait loci (QTLs) for berry firmness were identified on the basis of the genetic map and phenotypic data. Each QTL’s contribution rate varied from 21.5% to 28.6%. Four candidate genes that appeared to be associated with endoglucanase, ABA, and transcription factors were related to grape firmness. The most promising candidate genes for berry firmness were VIT_18s0041g02410 and VIT_18s0089g00210 [147].

For consumers of peaches, fruit firmness (FF) serves as a crucial predictor of fruit quality. This is why different researchers have looked into the genetic factors governing this property in peaches [73,117,148,149,150,151,152,153]. Fruit firmness has been proposed as the result of remodelling enzymes (including polygalacturonase (PG)) during ripening. Evidence suggests that this trait is under the control of a major gene (M/m) localised in linkage group 4 (LG4) [148]. In fact, in the position of the M locus, there are two genes coding for *endoPG*, one of which (Prupe.4G261900) has been proposed as a candidate for the determination of this trait [148,154]. EndoPG1 was associated significantly with low/high content of total sugars and high/low firmness. Both parameters are indirectly linked because when fruits are ripe, they have low firmness and high total sugar content [155]. The positions of the qtnFF 4.1 and qtnFF 4.2 (Table 1), which were reported by da Silva Linge et al. [33], matched the QTL interval linked to firmness loss that had been mapped by Serra et al. [152]. Moreover, the qtnFF 4.3 was located in the same genomic region where Zeballos et al. [117] and Carrasco-Valenzuela et al. [153] found a QTL for fruit firmness and a QTL for softening rate, respectively.

A major QTL for fruit firmness, named qP-FF4.1, which was previously unreported, was identified in three sweet cherry populations. In the sweet cherry germplasm, 13 haplotypes (alleles) associated with either soft or firm fruit were found, and the ‘soft’ alleles were dominant over the ‘firm’ alleles.

The fact that the majority of cherry cultivars are homozygous for the ‘firm’ alleles at qP-FF4.1 (Table 1) and that sweet cherry individuals with the ‘soft’ alleles are exclusively mazzards implies that this locus is a signal of selection. A gene called expansin was found to be the most promising candidate among those associated with plant cell wall remodelling and several hormone signalling pathways [156]. Additional minor QTLs on LGs 1, 2, 5, 6, and 8 were also detected [156]. Three QTLs, two on LG1 and one on LG6, were identified by Calle et al. [157]. The most significant were those on LG1 that were confirmed in two years, mapping to a nearby physical position, but their confidence intervals do not completely overlap and their QTL peaks are different [157]. Comparative transcriptome analysis of unripe and ripe fruits of banana, mango, guava, and papaya exhibited differential expression of genes associated with fruit softening [114]. However, no specific markers have been yet developed.

## 6. Organoleptic Qualities

### 6.1. Soluble Soil Content (SSC) and Sugar Content

Fruit sensory profiles, which determine overall quality as perceived by consumers, are made up of taste, aroma, and mouth-feel characteristics referred to collectively as ‘flavour’. The presence of water-soluble substances (such as sugars and organic acids) that contribute to a sweet or sour taste, as well as phenolic compounds that contribute to a bitter or astringent sensation, determine the flavour of fruit crops in large part. Sugars are a crucial part of what makes fruits edible, primarily adding sweetness, and are one of the main elements affecting how satisfied consumers are with the fruit as a whole. Both the overall sugar amount and the specific sugar profile affect the sweetness intensity. From an application standpoint, sugar content is typically determined by measuring the concentration of soluble solids (SSC or Brix). Brix degrees is a unit of measurement for the amount of dissolved solids in a liquid that can be determined with a refractometer. 

The contribution of other optically active compounds, such as pectins, salts, and organic acids, can cause SSC to vary. However, because the sugar content is significantly correlated with SSC [155,158], it is a reasonable surrogate measure of sugar content and the overall evaluation of fruit quality.

Concentrations of fructose, glucose, sucrose, and sorbitol have an impact on the sugar profile in various fruit crops [108,159]. However, environmental factors (such as light received, canopy position, water availability during fruit development, plant nutrition, thinning practices, and temperature during fruit development) also have a significant impact on this trait [160]. As a result, SSC is a complex trait that is also characterised by low inheritance. Due to this, the majority of scientific papers listed in the bibliography only discuss QTLs connected to the trait, which frequently account for less than 10% of phenotypic variation [44,161,162,163,164]. Determining the function of the discovered QTLs is frequently difficult as well because a variety of variables, including photosynthesis, metabolism, the capability of the sink organ, and others, affect sugar content and transport.

QTLs for sugar-related traits have been identified in strawberry, apple, sweet cherry, and apricot, indicating some synteny between these *Rosaceae* fruit crops. Using a bi-parental crossing of the apple varieties ‘Telamon’ and ‘Braeburn’, Kennis and colleagues [165] identified QTL clusters for SSC on LG2 and LG10. QTLs for SSC located at the bottom of LG2 co-localised also with the QTL for SSC detected by Guan et al. [166] using the Washington State University (WSU) RosBREED germplasm at three different evaluation points within 2 years of observations. Additionally, the authors conducted QTL studies on individual sugars (fructose, glucose, sucrose, and sorbitol). Individual sugar QTLs have been reported in several linkage groups. Among them, twenty QTLs for fructose, glucose, sucrose, and sorbitol were clustered on LG1 from 55 to 86 cM at three evaluation points in both years [166]. The proportion of variance described by the QTL in this area was especially high for fructose, accounting for 34–67% of phenotypic variation. Ma et al. [159] used an F1 segregating population obtained from the crossing of the cultivars ‘Jiguan’ and ‘Wangshanhong’ to identify a distinct cluster of QTLs in LG3 from 48,6 to 50,6 cM for the glucose, sucrose, fructose, and sorbitol content. An additional QTL for glucose content was detected on the LG4. These intriguing clusters identified by these studies could be caused by tightly linked genes for individual sugars or the pleiotropic effect of a single gene encoding a key enzyme involved in glycometabolism, such as an invertase that can convert sucrose into glucose and fructose, as suggested by Guan et al. [166]. A different class of genes could also be related to these QTLs. Recently, two SWEET family genes (*MdSWEET9b* and *MdSWEET15a*) were discovered on chromosomal areas with sugar content QTLs and are plausible candidates for regulating fruit sugar build-up in apples [167]. 

Similar-functioning genes have also been discovered in *Arabidopsis* [168,169]. *MdSWEET2e* is located on the same region of LG4 that harbours the QTL for glucose content detected by Ma et al. [159], while *MdSWEET15a* is placed in the region of LG16, which contains several QTLs for SSC and the amount of sorbitol and fructose detected by Kunihisa et al. [170] in Japanese apples. Evidence of the occurrence of this type of gene has also been discovered in peach. Zeballos et al. [117] identified two QTLs that control SSC character in LG4 and LG5. *SWEET* genes were annotated within these regions on the basis of the sequencing of the reference genome of cv. ‘Lovell’ [171]. The two QTL areas on peach LG4 and LG5 correlate to syntenic blocks on apple LG3 and LG6 [172], which contain QTLs for sugar content according to the studies mentioned above.

Several other authors have discovered QTLs that regulate sugar-related regions in peach in LG4 and LG5. Etienne et al. [161] employed an almond x peach F2 interspecific cross to uncover QTLs for SSC, glucose, fructose, and sucrose in LGs 4, 5, and 6. The LG4 QTL for SSC, glucose, and fructose levels were identified and linked with the ripening date and fruit development period. Font i Forcada et al. [155] found that the BPPCT015 marker was strongly related to sorbitol level, while CPPCT028 and endoPG1 were connected with total sugar content, according to an association analysis that helped identify these three molecular markers on LG4. Illa et al. [172] also described the relationship between the BPPCT015 marker and the sugar QTL in the peach. The LG5 genomic area was concerned by the QTL for sucrose level and also contained the D gene, a key dominant allele responsible for low acidity in honey-type peaches. Furthermore, the chromosomal region containing the marker FG209A was found on LG6, accounting for 37% of the variation in the amount of sucrose and 17% of the variation in SSC. With it being the codominant marker for FG209A, they were able to calculate the dominance/additivity statistic, which they calculated to be −1.6 for SSC and −0.6 for sucrose content, suggesting underdominance and partial underdominance, respectively [161]. PRUPeVp2, a vacuolar pyrophoshatase involved in the development of an electrochemical gradient across the tonoplast, was shown to be co-located with this QTL. As a result, it has been proposed that this gene could serve as a proton source for sugar transport across the tonoplast [161].

More recently, during the trial period between 2015 and 2016, Shi et al. [120] identified a total of 18 QTLs for SSC along LGs 1, 4, 5, and 6. Ten QTLs were found on the LG1, two on the LG4 and LG5, and four on the LG6, according to the authors. In the aforementioned 18 QTLs for SSC, 540 annotated candidate genes were annotated. The synthesis of secondary metabolites, the metabolism of fructose and mannose, and the metabolism of fatty acids are just a few of the activities that these candidate genes participate in. The qPCR analyses revealed that in the mature stages of fruits, SSC and fruit acidity content (FA) were positively correlated with the expression of two genes, *Prupe-1G284300* and *Prupe-6G307600* (UDP-glucose 6-dehydrogenase and glucose-6-phospate dehydrogenase, respectively). However, more research is needed to determine the precise role of these potential genes in the production of SSC and FA characteristics in peach fruit.

Rawandoozi and colleagues [173] performed a pedigree-based study on 162 seedlings from seven related F1 families produced from seven parents descending from 12 founders, allowing the mapping of one QTL for SSC (qSSC5) on LG5 at 60-72 cM that explained 17 to 39% of the phenotypic variance. The authors examined the SSC QTL region from a nucleotide sequence standpoint. Eight SNPs in the qSSC5 region were thus chosen for haplotyping. Six SNP haplotypes (H1-H6) were found across the seven parents, three of which were connected with the *Q*-allele (H1, H2, and H6) and three with the *q*-allele (H3–H5). More specifically, the *Q*-allele was related to a rise of 1.7°Brix and the AB haplotype of the pair of nearby SNP markers ss_600256 (14.6 Mb, 58.48 cM) and ss_600509 (14.9 Mb, 59.66 cM). Two peach parents, ‘TX2B136’ and ‘Galaxy’, were found to be QTL segregating. The H6 haplotype, which was inherited from ‘TX2B136’, had a stronger influence on raising SSC in peaches than others. A minor QTL was also found on the LG4 between the SNP markers ss_410794 and ss_414387, but unlike qSSC5, which was mapped consistently across, the minor QTL was environment-specific and was only found in the cv. ‘TX13’ [173]. qSSC5 and the QTL for titrable acidity (qTA5b) co-localised in the same genomic region. The fact that both QTLs had the same parent, TX2B136, as the source of their Q-allele and were in the same coupling phase lends additional major strength to the link between SSC and fruit acidity character. The co-localisation between qSSC5 and qTA5b may indicate that there is a single QTL with pleiotropic effects rather than two functionally independent but genetically linked QTLs. The SNP haplotypes of this novel QTL could be converted into a universal DNA test for both TA and SSC [173]. The quantitative trait nucleotides (QTNs) connected to SSC on the chromosome (Chr) discussed so far (Chr 1, 4, 5, and 6) were also highlighted in the recent GWAS analysis conducted by da Silva Linge et al. [33].

QTLs linked to the SSC feature have also been discovered in other *Rosaceae*. A general linear model was used to map ten SSC-associated loci on LG2, 3, 5, 7, and 8 in Chinese cherry [106]. The SSC-associated markers SAUCps302-255, SAUCps302-234, and SAUCps303-330 were discovered in the LG3 QTL called qP-TSS3.1m. In addition, the marker SAUCps303-330 was located around the peak association signal Glu.2013.3.1 for peach glucose content. As a result, Liu and colleagues speculated about the potential relevance of SAUCps303-330 as a functional marker for mining SSC-related genes, but more research is needed. 

A *Rosaceae* family-level strategy was utilised to uncover loci controlling SSC in blackberry, taking advantage of the synteny between the species [174]. This approach enabled the identification of three markers on Chr 1: BBS_SNP45, BBS_INDL31, and BBS_SNP46, which composed a QTL, named qSSC-Ruh-ch1.1, observed in three separate environments and accounted for a 1.5° Brix increase in SSC. All three alleles of the *qSSC-Ruh-ch1.1* gene were found in the introns that flanked the fifth exon of the gene marker Ro01-snap-gene-149.66 from the *R. occidentalis* v3.0 assembly. The genes linked to the QTL shared the most identity with sucrose synthase (*SUS*) genes from the glycosyltransferase-4 subfamily of the glycosyltransferase superfamily, according to the conserved domain and BLAST searches [174]. 

Finally, using a pedigree-based QTL analysis combined with the usage of the Axiom IStraw90 SNP array for genotyping, in strawberry, several QTLs for fruit quality including one QTL on the LG6A for SSC were validated [44]. The RosBREED strawberry population set used in this trais was screened using the SSR marker EMFv006, which was previously identified by Lerceteau-Köhler et al. [175] as being closely linked to the SSC QTL and was also discovered flanking the present QTL. Five different marker alleles were detected: 207 bp, 209 bp, 213 bp, 215 bp, and 217 bp. At the same time, the authors were able to confirm that there might be a significant association at the SNP level at the SSC locus on LG6A, which accounted for about 20% of the total SSC variation in the studied strawberry set, by visualising the QQ, Qq, and qq diplotypes through the FlexQTL platform and conducting analyses of the phenotypic distribution. The SSR marker EMFv006 alleles, however, were unable to distinguish between the FlexQTL-assigned QTL alleles. As a result, MAS applications were not appropriate for it [44].

SSC is a crucial trait in various species as well. Five QTLs for the SSC trait were found in the cashew apple using a mapping population. Three QTL (closet markers: 04CG035, 12CY062, and 05Bg131) were found in the male paternal CP96, a giant cashew genotype, and two QTL (closet markers: 07CY003 and 22Ag026) in the female parental CCP1001, a commercial dwarf cashew clone [163].

For clementine and mandarin fruit quality attributes, a genotypic BLUP model with additive and dominant effects was applied, along with a segregating population. The consensus map found one QTL for SSC on LG8 at 49,8 cM, which accounts for 19,8% of the total variance [108]. Another bi-parental population resulting from the ‘Fortune’ x ‘Murcott’ hybrid may have previously identified this QTL on LG8 [80]. Furthermore, QTLs for glucose and fructose showing a very high correlation rate (r = 0.95) were found on LG9, with both QTLs showing significant impacts (R2 > 30%) [108].

In grapevine, there was similarly a strong association between these two hexoses (r = 0.93) [176]. For three years, independent analyses of QTLs for SSC, specific sugars (glucose and fructose), and total sugar content were conducted. Because the QTL for total sugar and the QTLs for fructose and glucose overlap in this chromosomal region, one QTL in LG14 is potentially interesting in studies on grape breeding. Similar to this, the QTLs for fructose and glucose in LG14 overlapped with the QTL for SSC.

The use of different cross-populations and approaches permitted the identification of additional QTLs for the sugar-related characteristic in grape. One QTL for SSC was discovered on LG2 within a number of years of observation in a consensus map made using the mapping population 626-84 x Iku82, with a peak near the marker VCM6F1 [111]. QTLs were discovered for brix per cluster in LG1 and LG3, and in LG3, the SSR marker (VMC1a) accounted for 9% of the total phenotypic variance [177]. Researchers also discovered a negative association between this characteristic and titratable acidity. Moreover, within a table grape germplasm, a mixed linear model (MLM) was utilised to discover markers linked with genes affecting berry quality attributes. SSC was linked to three loci (Gret1Ra-890-8, VineFb-827-4, and VineFb-827-3) [109].

### 6.2. Pulp Acidity and pH

Some important organoleptic characteristics of fruit juices are formulated using measures of Brix (soluble solids) and titratable acidity (TA), as palatability is related to the sugar–acid ratio (Brix:TA) [178]. However, too little acidity can cause bland flavour, as an expression of fruit flavour requires a minimum level of acid, typically between 0.5 and 1.0% TA [178].

Using the OpenArray^®^ technology (Thermo Fisher Scientific, Waltham, MA, USA), a complete set of 128 apple SNPs connected to a variety of phenotypes addressed in apple breeding was converted into Taqman^®^ qPCR assays. Four markers on LG16 (AHFBAZU, AHHS7CA, AH89247, and AHBAIAO) (Table 1) were significantly linked (p 0.01) with the phenotypic data of acidity in the validation families out of 25 SNPs chosen for fruit acidity on LG8 and LG16. Of the four markers targeting the Ma1 locus of LG16 associated with fruit acidity, AHFBAZU was recommended, being the most strongly associated with the phenotype [67].

Malic acid and pH are linked characteristics, and similar QTLs in linkage groups LG1, LG2, and LG4 were found in apricot. PaCITA7 and UDAp-471 markers were flanking on either side of the LG1 QTLs (Table 1). Malic acid was closely correlated with PceGA34 and UDAp-473 (LOD = 5.37) markers during the course of two years in LG2, and the same QTL was discovered throughout the course of three years in LG4 with the support of UDAp-482, UDAp-439 (LOD = 4.83), and UDP96-003 markers. The QTL on LG2 mostly explained the difference in acidity levels [85]. Other authors have also described QTLs related to acidity in similar linkage groups in peach, mainly in LG2 [179,180] and in apricot [181].

Since it is connected to how consumers perceive sourness and sweetness on the basis of the sugar-acid ratio, which has a significant influence on consumer acceptability, pulp acidity is a crucial component of banana organoleptic quality [182]. pH is a useful indication for high-throughput phenotyping because it is able to accurately predict the sourness of the sensory qualities [183]. QTLs were found on LG1_7, 3, 4, 6, 9, and 10 for pulp pH. QTLs were found on LG1_7 of the ‘Pisang Madu’ cultivar in two different areas [144]. All fruit ripening days were affected by QTLs found in the centre of LG1_7 (pHWQTL-PM1-7.1: R2 = 19.3–50.6%, CI = 27.4 Mb). This central region of LG1_7 corresponded to the part of the reference chromosome 7 and the entire reference chromosome 1 that does not recombine in ‘Pisang Madu’. In the second region at the bottom of the LG1_7, corresponding to the banana reference chromosome 7, QTLs were detected for all fruit ripening days, except for days 0 and 6 (pHWQTL-PM1T7.2: R2 = 3.2–6.9%, CI = 2.2 Mb). These QTLs combined with those detected for pH reached a total R^2^ value of 67.8% for day 6 and 58.5% for day 7. On Pisang Madu’s genetic map, the QTL that contributed most to pulp acidity (R2 = 19.3–50.6%) was found on LG1_7, a parent that is closely related to Cavendish, the most widely cultivated group of dessert banana cultivars. The discovery of genes perhaps involved in pulp organic acid metabolism (*PEPC* and *PEPCK*, *Aco*, *GAD* and *Mal* activities, gene related to V-ATPase) was made possible by colocalisation between QTLs associated with pH and a list of candidate genes collected from the literature [144].

A recent study from Mengist et al. [184] described a high-density linkage map for blueberry and the first QTL mapping in tetraploid blueberry for fruit quality traits including pH and titrable acidity (TA). QTLs were identified for fruit quality traits including four QTLs for TA and two QTLs for pH. On chromosome 3, QTLs controlling TA and pH were uncovered. These QTLs are stable across time (3 years for pH and 3 years for TA), and they account for around 20% of the phenotypic variance for each trait. The QTLs have an opposite influence on the two characteristics, causing higher pH levels to result in decreasing TA values, as predicted by the correlation study. Another QTL-controlling TA was identified on chromosome 12 for the year 2018. This QTL explained 16% of the phenotypic variance and was associated with increasing TA values.

The sour, acidic flavour of *Vaccinium macrocarpon*, the American cranberry, is well-known. Although some acidity is necessary to express fruit flavour, cranberry products must contain a significant amount of ‘added sugars’ due to their high acidity. Two SSR markers within 1 cM were found using bulk segregant analysis employing simple sequence repeat (SSR) markers which co-segregate with the low citric acid phenotype. Genotypic variation revealed that the CITA locus, as was referred, was multi-allelic and had a hierarchy of semi-dominant alleles. Single-nucleotide polymorphisms (SNPs) were discovered through genotyping-by-sequencing for further fine mapping, which located the CITA gene on the distal end of chromosome 1. One marker (scf258d SSR) almost consistently (Table 1) co-segregated with the low citric acid phenotype [185]. In cranberry, also a relatively high concentration of malic acid contributes to the fruit acidity. For the mala locus, the KASP marker MA_476 (Table 1) had the best sensitivity and specificity for successfully determining lower MA individuals and, importantly, it could discriminate heterozygotes with the mala allele from homozygotes lacking the allele [185,186].

Titratable acidity (TA) is often associated with solid soluble content (SSC) in grapevine berry studies due to the importance of the SSC/TA ratio in the winery industry. Genetic control has been investigated, and several studies have reported the discovery of QTLs associated with these traits [187]. For the titratable acidity, QTLs were found across linkage groups 6, 13, and 19, and they explained low variation [176,188]. In a different study, one QTL for TA was detected on LG13 (in the genotype Iku82 and in the consensus maps) in 2008, 2009, 2011, and the 4-year average. Its peak was located at the top near marker VVIM63, VVS1, or VVIC51 (Table 1) [111]. In a complex grapevine hybrid population (*Vitis riparia* X ‘Seyval’), a single QTL for malic acid (MA) was found on LG06 with a peak at 70.24 cM (LOD = 6.24) [164]. This MA QTL explained 26.2% of the phenotypic variance and had a relatively small 1.8-LOD support interval spanning 7.3 to 8.4 Mbp. A single QTL was found on LG06 for the MA/SS ratio, peaking at 32.10 cM (LOD = 5.93). This MA/SS QTL accounted for 26.0% of the phenotypic variance and covered a rather small range between 3.5 and 4.3 Mbp. Positive additive effects suggested that ‘Seyval’ was the source of greater MA/SS ratios, and the dominant effect was minimal. Two QTLs that affected both traits were evident in the combined study of SS and MA. One QTL was found at LG01 at 49.94 cM (5.0 Mbp), while a second QTL at LG06 at 75.23 cM (8.3 Mbp) likewise exhibited strong evidence for simultaneously impacting both characteristics. When SS and MA were investigated independently, the study of the MA/SS ratio revealed a QTL on LG06 that was distant from the QTLs found for SS and MA. The region of grapevine orthologs of the aluminium-activated malate transporter ALMT1 covered the physical interval Chr06:7,985,435–11,876,418. This gene relates to the Ma1 and Ma2 loci in apples, which control the amount of malic acid transporters controlling the acidity level in apple fruits [164]. Razi et al. [109], in an association mapping study on Iranian table grape cultivars, found four loci associated with pH (Edel-836-1, Gret1Fa-857-8, Gret1Ra-890-4, Edel-855-3), and three loci associated with TA (Edel-836-1, Gentil-834-4, Vine1Fb-823-3).

The majority of studies on peach quality characteristics have largely focused on sugar content, assessed from soluble solid concentration (SSC), estimated in degrees Brix, and acid, measured as titratable acidity (TA) and/or pH, both of which affect the peach’s overall organoleptic quality [189]. This character is encoded by a main gene (*D/d*) on peach chromosome 5, where the non-acid character is controlled by the dominant D allele [190]. A single-nucleotide polymorphism (SNP) marker for this trait has been developed. Two-hundred and twelve accessions from 224 accessions tasting acid are TT. Two-hundred accessions from 212 accessions tasting non-acid are TC or CC (Table 1). The accuracy rate was 94.5% [191]. The authors created PCR primers based on the SNP marker to distinguish non-acid peaches. A total of 169 cultivars and 84 offspring were used to validate this method. Data were consistently reliable, and 80 progeny (95.2%) and 160 cultivars (94.7%) met the criteria [191]. In a previous study, a SSR marker (CPPCT040) was used in marker-assisted selection for the sub-acid trait (Table 1). This marker amplified six alleles from a sample of 231 cultivars, and the presence of one of them, the allele CPPCT040193, in homozygosis or heterozygosis, led to a TA lower than 5.5 g/L, which is consistent with the dominant character of the sub-acid trait [192]. Other authors identified a QTL associated with TA on LG5, and the mapping positions are consistent with markers developed for sub-acidic traits in peach [192] and with the co-localisation of QTLs for TA and pH on LG5 [117,180,193]. A range of 14 to 22% of the variance in the acidic phenotype was explained by a novel QTL for TA on the distal end of LG5. When the first QTL is homozygous for high acidity, this QTL co-localised with the QTL for SSC, and only in that situation affected TA (epistasis). The novel TA QTL (qTA5b) co-localised with the QTL for SSC that was found in this study (qSSC5). Because qSSC5 and qTA5b are co-localised, it would be possible that there is only one QTL with pleiotropic effects rather than two functionally distinct but genetically connected QTLs. A universal DNA test for both TA and SSC might be developed using the SNP haplotypes of this unique QTL [173]. Other studies using an association mapping approach confirmed QTLs collocated with the major locus for low-acid fruit (D-locus) previously reported [33].

Using the pedigree breeding method (PBA), many QTLs were found for horticulturally significant features in strawberries. Two QTLs for TA on LGs 2A and 5B, as well as a QTL for pH on LG4CII, were shown to affect fruit quality [44]. With regard to pH, a significant effect QTL was found on LG4CII, which accounted for 20% of the phenotypic variation. The phenotypic variation for TA ranged between 8 and 22%. Significant variations were detected often only in one parent. The pH QTL detected on LG4CII in 2012 was also previously reported on LG4 [162,175].

### 6.3. Dry Matter Content

The amount of photosynthates (sugar) and derivative compounds (such as starches, proteins, and structural carbohydrates) absorbed from the roots by nutrient uptake as well as the leaves during the photosynthesis process and transported to the fruit for storage and metabolism results in dry matter content (DMC). This metric appears to be a reliable quality index for a variety of tree fruit commodities. Indeed, DMC has been linked to positive consumer responses in European pears, as well as apple, cherry, bananas, and kiwifruit [194,195,196,197,198,199], because it contains many of the physio-chemical attributes that contribute to the sensory perception of texture, sweetness, and flavour. Furthermore, consumers turn out to be willing to pay more for high-DMC pears [194] and are more likely to purchase high-DMC apple and kiwi than lower-DMC varieties [196,200]. Finally, DMC changes throughout the fruit’s development, and this correlation makes it a good indicator of maturity and fruit quality in both climacteric and non-climacteric fruits. Producers and the fruit industry are therefore increasingly interested in selecting high-DMC fruits in order to efficiently achieve higher-quality yields. However, the accumulation of dry matter in the fruit is also the result of resource allocation within the canopy, which is influenced by the interaction of numerous variables, including growth potential (genetics), resource availability (environment), and inter-organ competition fuelled by source–sink relationships. As a result, it is very challenging to associate reliable molecular markers that can differentiate between high and low DMC.

In apple, a segregating population was used to assess the contribution of genetic makeup and growing environment on fruit maturation and fruit quality (DMC and firmness). Chagné et al. [201] phenotyped for fruit maturation timing, firmness, and DMC in three climatically diverse apple-growing regions in New Zealand over two seasons. For apple DMC, 30 QTLs were found and located on ten LGs, accounting for 3% to 21.8% of the phenotypic variation. The authors observed that there was an interdependency of fruit maturation date, DMC, and storage potential within this population for those loci that were environmentally stable over three sites, with two regions on LG10 and 16 strongly contributing. These loci would unintentionally result in selection for high DMC and later maturing apples if they were used in a MAS selection program to choose progeny bearing firmer fruit [201].

DMC was also detected in a banana segregating population genotyped by sequencing over a 7-day ripening period and three production cycles. QTLs were found on LG5, 6, 8, 9, 10, and 11 for DMC. On days 1, 3, and 7, the total R^2^ values were 60.4%, 31.4%, and 59.1%, respectively. These values correspond to five, two, and five different QTLs, respectively [144]. On the ‘Pisang Madu’ (banana cultivar closely related to ‘Cavendish’) map, two QTLs with significant effects were discovered near the top of LG9. Enzymes involved in glycolysis, sugar metabolism, transport, and specific hormonal activity are potential candidate genes for DMC accumulation according to their functions. In this case, two candidate genes involved in the cell wall, which is the structural component of the DMC, were discovered within the two QTLs on the LG9 [144]. One significant finding is that DMC demonstrated high genetic determinism in the studied progeny (61.3% PVE), supporting the potential of this trait for breeding programs.

### 6.4. Volatile Organic Compounds

As stated in the previous sections, fruit sensory profiles play an active part in the establishment of flavour, which is one of the most significant attributes of the fruits, as well as other factors such as appearance, texture, and nutritional contents. Along with sugars and acids, volatile organic compounds (VOCs) are a different category of products that play a crucial role in determining the flavour of fruits and have a large influence on how people perceive and accept them. The blend of volatiles includes alcohols, aldehydes, ketones, sesquiterpenes, phenylpropanoids, and esters, which are produced from primary metabolites. More than 1700 VOCs have been found to exist in 90 distinct plant families [202].

In the context of using molecular markers in VOCs research, a full-sib parental mapping population was used to conduct a QTL survey to evaluate the VOC segregation found in apple fruit collected after a two-month postharvest storage period using a novel proton transfer reaction time of flight mass spectrometry (PTR-ToF-MS) [203]. A group of QTLs distributed over ten chromosomes in the population, such as linkage groups LG2, 3, 4, 5, 11, 13, 14, 15, and 16, were identified through the combined analysis of markers and phenotypic data. The QTLs found in these areas had LOD values ranging from 3.52 to 14.2, and their respective expressed variance percentages ranged from 26% to 68.9%. The majority of the detected QTLs were linked to sesquiterpenes, alcohols, and esters, according to the annotation of the masses associated with the VOCs profiled by PTR-ToF-MS [203]. Indeed, among all the compounds, esters are the largest class of VOCs in apple aroma and are produced either through the isoleucine pathway or the fatty acid pathway [204]. It is noteworthy that a functional marker (Md-AAT1 SSR) designed for the *MD-AAT1* gene (*MDP0000214714*), which is involved in the final step of the ester biosynthetic pathway, is co-located with the set of QTLs found on LG2 that are primarily associated with esters. When screening a collection of 124 apple accessions, this SSR marker, which is situated 36.2 kb downstream of the *MD-AAT1* gene, allows for the detection of eight alleles (119, 201, 203, 210, 212, 216, 218, and 226). The allele 201 and major VOCs were found to be significantly correlated by the candidate gene association analysis. In contrast to acetate esters, which showed a trend in the opposite direction, the presence of allele 201 generally determined a significant increase in the production of volatiles [203]. However, Larsen et al. [205] discovered convincing associations for acetate esters, particularly butyl acetate and hexyl acetate, on chromosome 2 in the region, including the *AAT1* gene, using GWA studies. SNPs spread across a two- to three-Mb region were found to have significant associations with these VOC compounds. The authors reported that for butyl acetate and hexyl acetate, heterozygous cultivars (CT and AG, respectively) for the loci chr2:1,258,734 and chr2:1,730,413 had approximately five times higher ester contents than homozygous cultivars with genotypes CC and GG, respectively. Differences in VOC composition observed between the two works could be attributed to the fact that Larsen and co-authors [205] measured VOC content in juice samples, which may differ from the volatile composition of whole apples. When apples were crushed, a rapid enzymatic process began the hydrolysis of some esters, while the content of other esters such as butyl acetate and hexyl acetate rapidly increased. In both cases, a critical “hot spot” for genes involved in ester biosynthesis has been proposed to be at the top end of chromosome 2.

The strawberry market is interested in improving varieties in terms of flavour as well. In this case, the volatile profiles include esters, ketones, terpenes, furanones, aldehydes, alcohols, and sulphur-containing compounds. One important VOC contributing to flavour is γ-decalactone (γ-D), which provides a ‘fruity’, ‘sweet’, or ‘peachy’ aroma. A distinct feature of this compound is that it is prone to being undetectable in some genotypes relative to others, for which accumulation can differ significantly within and between harvest seasons [206,207]. On LG3 (LG3:31,112,418–31,114,643), a single gene (gene 24414), designated as *FaFAD1*, was found to segregate with the presence of the γ-D volatile as a single dominant locus [208]. Using *FaFAD1* primers within the 5′ of the gene, a γ-D PCR-based assay was developed by the authors and tested against three populations (one F1 population derived from the cross between ‘Elyana’ and ‘Mara des Bois’, a cultivar where γ-D has not been detected, and two BC1 populations), as well as a group of cultivars with demonstrated present or undetectable γ-D. The 500 bp PCR amplicon was only found in genotypes that have been shown to produce γ-D in all individuals. Additionally, an SSR marker was created to look into the cosegregation of alleles located more distantly from *FaFAD1*. The *FaFAD1* gene is 11 kb away from the SSR sequence, and polymorphisms associated with the SSR sequence showed segregation with the *FaFAD1* gene. ‘Elyana’ displayed four marker alleles (205, 209, 215, and 219), whereas ‘Mara del Bois’ only has the 209 allele. While alleles 215 and 219 were not found to be associated with the -D phenotype, all genotypes tested for allele 209 were found to be monomorphic, whereas allele 205 is a trustworthy indicator of the ability to produce γ-D [208]. Noh and co-workers [209] developed a high-throughput marker-assisted selection system that combines rapid DNA extraction, high-resolution melting, and SSR analysis for selection at the *FaFAD1* locus (Table 1), which confers a peach-like aroma.

VOCs were investigated also in mandarin using a bi-parental population derived from the cross between cv. ‘Fortune’ and cv. ‘Murcott’ during two harvest seasons. A total of 206 QTLs were identified for 94 volatile compounds, including 17 aroma active compounds; among them, 25 were consistent over multiple harvest times. A genomic region containing the genes encoding geranyl diphoshate synthase 1 (*GPS1*), terpene synthase 3 (*TPS3*), terpene synthase 4 (*TPS4*), and terpene synthase 14 (*TPS14*) corresponds to a QTL interval controlling multiple monoterpenes and sesquiterpenes on ‘Murcott’ LG2, in particular [210]. Although validations on larger breeding germplasm and segregating populations are ongoing, all 206 QTLs were verified using 13 citrus selections, and some of them showed the potential for MAS in citrus breeding programs (data not provided by the authors). After further validation, these SNPs could be utilised to screen mandarin individuals for particular volatile compounds or even aroma active compounds [210].

A major QTL for linalool, another monoterpene compound, was also detected in peach in the distal part of LG4. A major QTL for nonanal, an aldehyde reported as an odour-active compound concentration, is located in the same genomic region. Finding seedlings that lack or produce linalool and have higher or lower concentrations of nonanal may be made easier by choosing the right allelic configurations of molecular markers at the distal region of linkage group 4. These findings may present opportunities for aroma MAS in peaches, but more investigation is needed to determine the significance of linalool, nonanal in the perception of peach aroma [211]. Doligez et al. [212] identified a major QTL on linkage group (LG) 5 associated with linalool in grapevine using simple sequence repeat (SSR) markers. The QTL was also linked to two other VOCs (nerol and geraniol) and was discovered within an 8,3 cM interval between markers VRZAG79 and VVC640. Other authors [213,214] who worked on different populations confirmed the location of this QTL on LG5. The candidate gene was identified as a *V. vinifera* 1-deoxy-D-xylulose-5-phosphate synthase (*VviDXS*) gene due to its co-localisation with the LG5 QTL associated with monoterpene content [213]. The *VviDXS*-associated QTL explained 17-93% of the variation in linalool, nerol, and geraniol concentrations.

## 7. Harvesting Date, Maturity Date, Ethylene, Ripening

The climacteric fleshy fruit ripening process is known to be regulated by three main types of transcriptional circuits. The ripening circuits were formed by duplicated MADS transcription factors in eudicots with recent WGD (apple), as opposed to NAC transcription factors in eudicots without recent WGD (peach). Bananas, a monocot plant, also underwent recent WGD and have two coupled circuits formed by the MADS and NAC genes. It is possible that these independently evolved ripening mechanisms descended from pre-existing pathways that served various functions in the ancestor’s angiosperms because the ripening genes, as well as their epigenetic marks restricting their expression, are conserved in their orthologues in non-climacteric fruits and even dry fruit [215].

Harvest date, eating time, and earliness in apple were in strong SNP association with a NAC transcription factor gene and sequencing identified two haplotypes associated with harvest date (Table 1) [205]. The same three SNPs on chromosome 3, which are only separated by 14 and 104 base pairs, showed substantial relationships with each of the three traits [32,216,217]. The SNP chr3:31409362 is a nonsynonymous mutation in the gene *MDP0000868419* (*MD03G1222600* in GDDR13) encoding a NAC transcription factor. Using NAC sequences annotated from wild strawberries (*F. vesca* L.), Moyano et al. [218] recently examined the expression of NAC transcription factors during fruit ripening in strawberry (*Fragaria’ananassa* Duchesne ex Rozier) and revealed that *FaNAC035*, the closest homolog to apple *NAC18*, is expressed in fruits during ripening. Banana (*Musa acuminata* Colla), tomato (*Solanum lycopersicum* L.), and peach (*Prunus persica* (L.) Batsch) fruit ripening has also been related to *NAC TFs* [219,220,221].

On LG11 in a 4-year average, one QTL for harvest date in grape was found. Despite the fact that the position of its summit changed over the years, it was situated in the upper to the middle section of LG11 [111].

Nishio et al. [222] identified four loci (LG-A, -D, -F, and -H) across the two populations of Chestnut QTLs for HARVEST were identified in the middle of LG-D. The most significant QTLs (qtl-HARVESTKu-D and qtl-HARVEST-T43-D) were detected in both populations.

A QTL with a significant correlation to harvest time was discovered at the top of LG15, next to the PPACS2 locus in Japanese pear (*Pyrus pyrifolia* Nakai). Two *ACC* synthase genes, *MdACS1* [223] and *MdACS3* [224], were found to be mapped in LG15 of the apple, and Kunihisa et al. [170] found that the harvest time QTL at the bottom of LG15 around *MdACS1* was strongly associated with preharvest fruit drop in the apple. Both Japanese pear and apple *ACC* synthase genes (*PPACS2*, *MdACS1*) have various functions regulating harvest time, preharvest fruit drop, and fruit storage capability [81].

Genes involved in maturation time and flesh firmness were consistently mapped to three regions of the peach genome, which were discovered on LG4, LG5, and LG6. All included a QTL that accounted for a large portion of the variation in the maturity date character, and all were previously described for peach and other Prunus species. The major QTL in the central part of LG4 (qP-MD4) has been repeatedly reported by other authors in peach [73,220,225] and in other Prunus such as almond, cherry, and apricot [95,226,227]. It has been identified as a single major gene (*MD/md*) on G4 in some crossings [73,220]. The slow ripening (*Sr/sr*) gene is located in this same region, and it has been proposed that the *sr* allele, which determines the SR character, is one allele of MD [151]. In a peach F2 population, Dirlewanger et al. [226] also discovered the QTL on G6 (qP-MD6), which accounts for a significant fraction (15.7–30.2%) of the phenotypic variability for this trait. The aggregation of QTLs for several fruit features (fruit weight, skin colour, soluble solids content, acidity) on the centre region of G4 was attributed by Eduardo et al. [73] to a pleiotropic effect of the *MD* gene. Additionally, Nuez-Lillo et al. [151] discovered that the sr allele that determines this character co-maps with a large deletion at the position of the *Prupe.4G186800 NAC* transcription factor gene (Table 1) that Pirona et al. [220] identified as a potential candidate for MD. Eduardo et al. [149] also discovered the position of the Sr gene, which determines fruits that never ripen or soften, at the same place as *MD* [152]. The slow ripening (SR) feature in peaches is a mutation that inhibits the fruit from ripening normally. The recessive homozygote (*sr/sr*) of this gene, which has been localised to linkage group 4 of the peach genome, confers the SR phenotype and is responsible for determining this trait. It has been suggested that the causative mutation is a massive 26.6 kb deletion that includes the sequence of a transcription factor for the NAC (Table 1). A codominant molecular marker co-segregating with the *SR* phenotype was developed, allowing for the discrimination of two DNA fragments of different sizes associated with normal-ripening alleles, in addition to a third fragment associated with the *sr* allele (Table 1) [228].

In Japanese plum, a QTL was strongly localised in LG4 between 6 and 12 Mbp. The occurrence of this QTL is highly conserved within Prunus species, being related the MD locus with NAC family transcription factors [68].

QTLs controlling the ripening phases in raspberry were found across four chromosomes, namely, 2, 3, 5, and 6. Each of the groups had markers that had a significant influence at various ripening stages. In contrast to the Hh genotype of Gene *H*, which was linked to a slower rate of ripening across all phases from open flowers to the green/red stage, the presence of the AFLP marker P12M121-194 on chromosome 2 is connected with a shorter time to red fruit (Table 1). A wide variety of significant markers found on chromosome 3 that strongly influenced ripening suggest that more than one QTL was involved. There are two distinct markers on chromosome 5—ERubLR SQ01 M20, an EST related to floral development, and *RiMADS 01*, a *MADS* box gene that significantly influenced the rate of ripening. According to Danielevskaya et al. [229], these genes control the transition from vegetative to reproductive development, as well as the identification of the flower meristem and flower organs [230].

## 8. Conclusions—Perspectives

There is little doubt that genomic technology will hasten the creation of new cultivars of a variety of fruit crops, influencing the process of choosing genetically elite progeny for development as prospective new commercial cultivars as well as for use as parents in the following stage of breeding. With the use of next-generation sequencing, GWS can now be used to screen individuals who carry a significant number of SNP markers that are densely scattered throughout the genome. The same methods can also be used to find allele-specific markers for foreground MAS application before GWS. Since reliable phenotyping is a precondition for both the training of a GWS model and for genome-wide association studies, such activities call for strong collaboration between researchers from several fields. Although the search for molecular markers associated with agronomic traits has assumed an important role in breeding programs of fruit species, it frequently seems necessary to validate the results obtained in a greater number of varieties and genotypes. There are 44 molecular markers immediately available to the community of breeders, for 9 characters, and they are usable in 15 different fruit species or genera. However, it is evident that there is the need to deepen the studies relating to the many QTLs identified in order to be able to obtain useful markers for an efficient assisted selection. For many important traits, with complex genetic control, for example, fruit-soluble solid content or dry matter, important for fruit shelf life, we still do not have reliable markers, despite the considerable work done in many species.

## Figures and Tables

**Figure 1 ijms-24-08984-f001:**
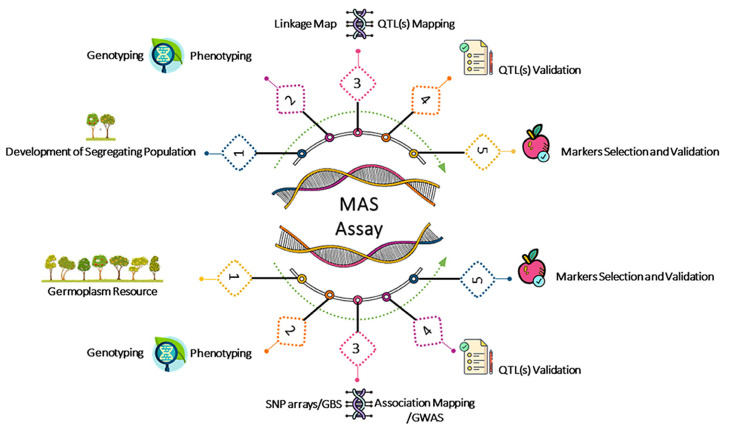
Schematic workflow for marker assay development.

**Figure 2 ijms-24-08984-f002:**
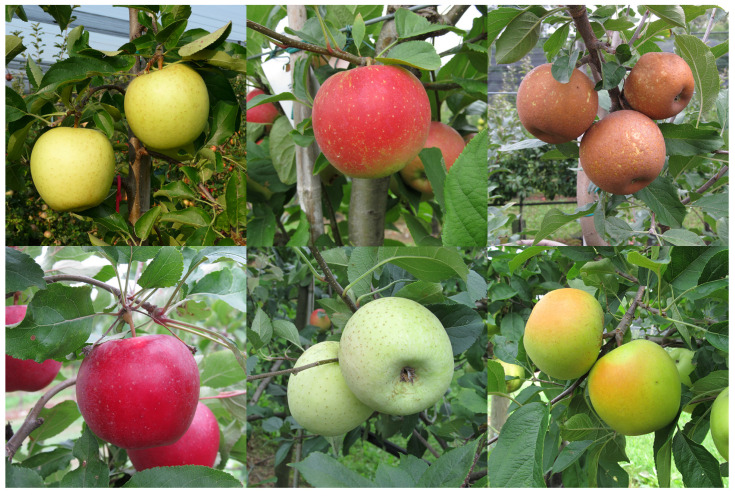
Illustrative panel of different types of apple skin colours. All the apple fruits shown are selection lines from the University of Udine’s breeding program.

**Figure 3 ijms-24-08984-f003:**
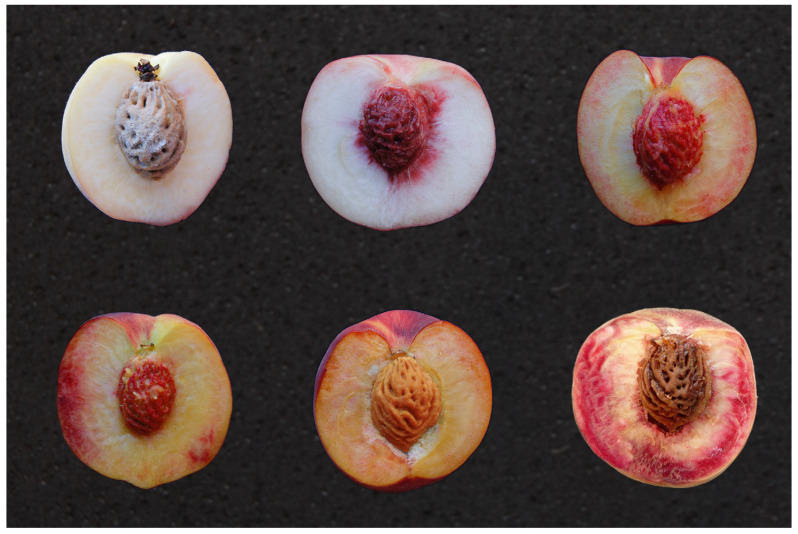
Illustration showing various shades of peach flesh. The peach fruits displayed are all the result of earlier research conducted at the University of Udine.

**Table 1 ijms-24-08984-t001:** List of the most relevant marker for MAS breeding.

Character	Phenotype	Species	Marker	Type
Phenological traits related to fruit production	Flowering time	Apricot	UDAp-423r, AMPA-105	SSR
	Pollen sterility	Peach	*Ps*	Gene
	Sex determination	Ginco	GBA, GBB	SCAR
	Sex determination	Kaki	DlSx-AF4S	SCAR
	Sex determination	Kiwifruit	*SyGI*, *FrBy*, *Ank*	Gene
	Sex determination	Mulberry	MBS markers	RAD-seq
	Sex determination	Papaya	SCART1, SCART12, SCARW11	SCAR
	Sex determination	Pistacio	SNP-PIS-167992, P-ATL-91951–565	KASP
Skin colour	Red skin colour	Apple	Mdo.chr9.4	SSR
	Red skin colour	Apple	ss475879531	SNP
	Red skin colour	Peach	PpMYB10.1	Indel polymorphism
	Red skin colour	Strawberry	18bp-del-5’UTR-ANR	Indel polymorphism
Flesh colour	Dark purple-red/yellow colour	Cherry	LG3_13.146	SSR
	Red/yellow colour	Papaya	CPFC1	Indel polymorphism
	White/yellow colour	Peach	ccd4-INS1-f/ ccd4-E2-r	Indel polymorphism
Fruit size and weight	Increase fruit size	Sweet cherry	PavCNR12, PavCNR20	SSR
	Increase weight	Grapevine	VVIB19	SSR
	Increase weight	Japanese plum	UDAp456	SSR
Seedlessness	No seed	Citrus	OPAI11-0.8, OPAJ19-1.0, OPM06r-0.85, OPAJ04r-0.6	RAPD
	No seed	Citrus	AFLP-2, AFLP-5	SCAR
	No seed	Grapevine	UBC-269500	RAPD
	No seed	Grapevine	SCC8	SCAR
	No seed	Grapevine	SCF27	SCAR
	No seed	Grapevine	VMC7F2	SSR
	No seed	Grapevine	*VviAGL11*	Gene
	No seed	Grapevine	chrG8:11240814	SNP
Firmness, crispness, texture	Firmness loss	Apple	Md-PG1	SNP
	Firmness loss	Apple	*NAC18.1*	Gene
	High-texture performance	Apple	Md-PG1SSR	SSR
	Firmness loss	Grapevine	VMC2E7	SSR
	Firmness loss	Sweet cherry	qtnFF 4.1, qtnFF 4.2	SNP
Pulp acidity and pH	Acidity	Apple	AHFBAZU, AHHS7CA, AH89247, AHBAIAO	SNP
	pH and malic acid	Apricot	PaCITA7, UDAp-471	SSR
	Acidity	Cranberry	scf258d SSR	SSR
	Malic acid	Cranberry	MA_476	KASP
	Titratable acidity	Grapevine	VVIM63, VVS1, VVIC51	SSR
	Titratable acidity (acid/non-acid)	Peach	D_allele_SNP	SNP
	Titratable acidity (acid/non-acid)	Peach	CPPCT040	SSR
VOCs	γ-Decalactone flavor	Strawberry	FaFAD1	SSR
Harvesting date, maturity date, ethylene, ripening	Harvesting date	Apple	chr3:31409362	SNP
	Slow ripening	Peach	del_Prupe.4G186800	Indel polymorphism
	Slow ripening	Raspberry	P12M121-194	AFLP

**Table 2 ijms-24-08984-t002:** Size and weight QTL identified in fruit species.

Species	Trait	Mapping Strategy	Linkage Group	Candidate Gene(s)	References
Sweet cherry	Size	BP	2	*FW2.2/CNR*	[105]
Chinese cherry	Size	AM	8	Auxin response, cell differentiation, pectin biosynthesis	[106]
Chinese chestnut	Nut weight, width, thickness, height	BP	1		[107]
Citrus	Size, weight, diameter	BP, AM	2, 4, 5, 7, 8		[79,108]
Coffee	Yield, bean size	BP	2, 4, 6		[45]
Grape	Weight	AM, BP	1, 8,11, 17, 18	*Aux/IAA9*, DELLA protein	[109,110,111]
Japanese plum	Weight	BP	2		[112]
Logan	Weight	BP	1, 4, 10, 14	*FW2.2/CNR*, *P450, EXP4*	[113]
Peach	Weight, diameter	BP, AM	1, 2, 3, 4, 5, 6, 7		[33,115,116,117,118,119,120]
Guava	Weight, size, yield				[114]
Papaya	Weight, size				[114]
Banana	Weight		3		[114]
Walnut	Weight, size	AM	1, 3, 6, 8, 12, 5, 11, 15, 7, 9, 14	Beta-galactosidase, *RBK1*, *BEL1*-like	[121]

BP = bi-parental mapping; AM = association mapping.

## Data Availability

No new data were created or analyzed in this study. Data sharing is not applicable to this article.

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
