# Peer review of "Marker-Assisted Selection in Breeding for Fruit Trait Improvement: A Review"

_ijms, 2023, doi:10.3390/ijms24108984_

Round 1

Reviewer 1 Report

This manuscript entitled ‘‘Marker assisted selection in breeding for fruit traits improvement: a review"; could be good for publication in IJMS (ISSN 1422-0067).

This may be interesting, but some important points need to be resolved. Importantly, a study must provide a critical analysis of the data. In other words, you must assess whether specific data published really stand up to scientific scrutiny. In order to achieve the above, you must clearly define your specific aims and objectives. So in your study you must develop a critical appraisal of the state of the art. This is an essential element of any article. There are important scientific questions (both conceptual and methodological) which need to be addressed with the primary studies. A study must highlight this. The introduction, which is written in clear language, covers a number of relevant issues. Information are noteworthy, and not are correct supported by similar results from the specialty (WOS: 000339050700030, WOS: 000327818000032, WOS: 000318221200014,  WOS: 000229981900029). Try to rewrite the abstract and conclusions, I also recommend the nuance of the introduction, the way of working is not very well explained, the procedure is tedious and unsustainable. For this reason, I recommend that the authors try to use more sustainable methodologies, the interpretation of the results can be improved/ reformulated,

 Moderate editing of English language

Author Response

The authors thank the editor and kind colleagues who reviewed the manuscript. Our goal was to gather the essential information regarding the research and use of molecular markers useful for the selection process, using our direct knowledge and those obtained from the bibliography, specifically focusing on fruit species. In fact, our research group deals with the improvement of fruit species. From our direct interest in the topic, it was easy for us to propose a review of these research topics. We are aware that our effort may appear to be oriented more towards some species but we have tried to be neutral and transversal by collecting most of the works published on the subject. We certainly have to apologize to the authors who will not find their works cited. We have made a weighted choice trying to collect the most relevant works, describing the results obtained and we have tried to minimize the length of the manuscript which, in any case, is relevant. We make an attempt to address the three reviewers' conclusions. Below, we try to respond individually to the findings of the three reviewers.

Review 1

We define our specific aims and objectives in the introduction section including a specific paragraph, as suggested (lines 43-55).

In the last paragraph of the introduction section (lines 196-201), the methodology of the review is outlined.

In the supplementary material we summarize, for the convenience of the readers, the most relevant references.

In the conclusion section we emphasized the most significant results regarding the molecular markers that can be effectively used in marker-assisted selection to improve and accelerate breeding programs. All the changes are highlighted in red within the text.

Reviewer 2 Report

The research presented in the manuscript the development of marker assisted selection in breeding for fruit traits, including RAPD, SSR, SCAR, SNP, and GWS. According to the aim of the research, the authors red a large number of literatures and summarized well.

The paper contained good views based on many previous reports which were very well selected by authors. Also, the main views obtained by the authors are summed up, exhibiting a careful work.

To sum up, I find this paper very interesting and I think that the review is suitable for publishing in “International Journal of Molecular Sciences”.

English language is good.

Author Response

The authors thank the editor and kind colleagues who reviewed the manuscript. Our goal was to gather the essential information regarding the research and use of molecular markers useful for the selection process, using our direct knowledge and those obtained from the bibliography, specifically focusing on fruit species. In fact, our research group deals with the improvement of fruit species. From our direct interest in the topic, it was easy for us to propose a review of these research topics. We are aware that our effort may appear to be oriented more towards some species but we have tried to be neutral and transversal by collecting most of the works published on the subject. We certainly have to apologize to the authors who will not find their works cited. We have made a weighted choice trying to collect the most relevant works, describing the results obtained and we have tried to minimize the length of the manuscript which, in any case, is relevant. We make an attempt to address the three reviewers' conclusions. Below, we try to respond individually to the findings of the three reviewers.

Review 2

We thank the reviewer for his appreciation of the work we have done.

Reviewer 3 Report

Indeed, in general plant breeding is a difficult task, to worsen the situation breeding of fruit species is even tougher job due to breeding cycles and cost of breeding. Hence, very few fruit or tree species are attempted for trait improvement. In some fruit crops and horticultural crops, vegetative propagation is successfully used, however, there are limitations of vegetative breeding, as heterosis is not possible. In this manuscript, the authors have analyzed application of molecular markers to hasten breeding of fruit crops and horticultural trees.

The article is timely review of molecular methods and it covers most interesting areas of the breeding of fruit crops. The article could be considered for publication after careful revision.

Some comments:

·         Some successful examples could be included in abstract.

·         Line 64: Please describe acronyms prior to usage, e.g. SSC.

·         Some paragraphs L190-236 are two lengthy. It could be shortened. Same should be followed for other big paragraphs.

·         If possible, specific references could be added for Table 1 and Table 2.

·         Figure 2. Please give credit to the figures. If available, author could provide photographs of other important fruit crops.

·         The scientific literature review needs to be updated, for instance the introduction and discussion can be enriched with inclusion of additional studies on horticultural crops. I recommend authors use some other studies on DNA-based molecular markers to improve the scientific literature. See below few examples from literature on molecular marker-based analysis (using RAPD, RFLP, AFLP, SSR, ISSR, ITS, etc): Multiplex molecular marker-assisted analysis of significant pathogens of cotton (Gossypium sp.), 2022; Biocatalysis and Agricultural Biotechnology https://doi.org/10.1016/j.bcab.2022.102557 (Cotton);  Assessment of genetic diversity and volatile content of commercially grown banana (Musa spp.) cultivars, Hinge et al., Scientific Reports, 2022; https://doi.org/10.1038/s41598-022-11992-1 (Banana); Microsatellite and RAPD analysis of grape (Vitis spp.) accessions and identification of duplicates/misnomers in germplasm collection, Upadhyay et al., 2010 Indian J Hortic Volume 67 Pages 8-15; Microsatellite analysis to differentiate clones of Thompson seedless grapevine, Upadhyay et al., 2010, Ind Journal of Horticulture, Volume 67 Issue 2 Pages 260-263.

L1297: Change it to “Conclusions and perspectives”

Language and style is okay.

Author Response

The authors thank the editor and kind colleagues who reviewed the manuscript. Our goal was to gather the essential information regarding the research and use of molecular markers useful for the selection process, using our direct knowledge and those obtained from the bibliography, specifically focusing on fruit species. In fact, our research group deals with the improvement of fruit species. From our direct interest in the topic, it was easy for us to propose a review of these research topics. We are aware that our effort may appear to be oriented more towards some species but we have tried to be neutral and transversal by collecting most of the works published on the subject. We certainly have to apologize to the authors who will not find their works cited. We have made a weighted choice trying to collect the most relevant works, describing the results obtained and we have tried to minimize the length of the manuscript which, in any case, is relevant. We make an attempt to address the three reviewers' conclusions. Below, we try to respond individually to the findings of the three reviewers.

Review 3

We included examples in the abstract (line 24-27). All the changes are highlighted in red within the text.

We define the acronyms in line 80.

We have tried to be as short as possible in describing the huge number of results obtained in the scientific community. We carefully select a number of important references to be included in our review for each trait.

The references related to the information in Table 1 are reported in the supplementary material Table S1 to avoid having too large tables in the main text. We have added the references related to the information in Table 2.

We have included another example in the flesh color section for peach, Figure 3. Many examples could be included but we decided to use only original photos of the authors or of the research team.

We appreciate the reviewer for providing the list of interesting scientific works. After carefully reading these works, however, we made the decision not to include them in the review because the focus of these studies is more on molecular marker-based fingerprinting and germplasm analysis, which is not the purpose of the review. In fact, the work is focused on fruits and the molecular markers available within different species for marker-assisted selection for various traits linked with fruit quality, as stated in the integrated section of the introduction.

Round 2

Reviewer 1 Report

Accept in present form

Accept in present form